# NEURAL ADDITIVE TENSOR DECOMPOSITION FOR SPARSE TENSORS

## ABSTRACT

Canonical Polyadic Decomposition (CPD) is a fundamental technique for tensor analysis, discovering underlying multi-linear structures represented as rank-one tensors (components). The simplicity of the rank-one tensors facilitates the interpretation of hidden structures within tensors compared to other types of conventional tensor decomposition models. However, CPD has limitations in modeling nonlinear structures present in real-world tensors. Recent tensor decomposition models combined with neural networks have shown superior performance in tensor completion tasks compared to multi-linear tensor models. Nevertheless, one drawback of those nonlinear tensor models is the lack of interpretability since their black-box approaches entangle all interactions between latent components, unlike CPD, which handles the components individually as rank-one tensors.

To overcome this major limitation and bridge the gap between CPD and various state-of-the-art neural tensor models, we propose NEURAL ADDITIVE TENSOR DECOMPOSITION (NEAT) to accurately capture non-linear interactions in sparse tensors while respecting the separation of distinct components in a similar vein as CPD. The main idea is to neuralize each component to model non-linear interactions within each component separately. This not only captures the non-linear interactions but also makes the decomposition results easy to interpret by being as close to the CPD model as possible. Extensive experiments with six large-scale real-world datasets demonstrate that NEAT is more accurate than the state-of-the-art neural tensor models and easy to interpret latent patterns. In the link prediction task, NEAT outperforms CPD by 10% and the second-best performing neural tensor model by 4%, in terms of AUC score. Finally, we demonstrate the interpretability of NEAT by visualizing and analyzing latent components from real data.

## 1 INTRODUCTION

A tensor is a natural way to represent higher-order interactions between multi-aspect data. Tensor decomposition is a fundamental method for analyzing tensors by extracting latent structures as a set of factor matrices. Among the well-known approaches, Canonical Polyadic Decomposition (CPD) (Carroll & Chang, 1970; Harshman et al., 1970) has gained popularity due to its simplicity and ability to uniquely identify the latent components of the tensor (Sidiropoulos & Bro, 2000; ten Berge, 2000; Kolda & Bader, 2009) and has been central to a diverse range of applications such as healthcare analysis (Ho et al., 2014; Afshar et al., 2021), social networks analysis (Papalexakis et al., 2013; Al-Sayouri et al., 2020), knowledge base completion (Lacroix et al., 2018; 2020) and recommendation (Yao et al., 2015; Chen & Li, 2020).

CPD stands out for its interpretability to facilitate understanding of latent patterns with its simple structure among different types of tensor decomposition models (e.g., Tucker) (Kolda & Bader, 2009; Papalexakis et al., 2016). As depicted in Figure 1(a), CPD reconstructs a tensor as a sum of rank-one tensors, where each represents a unique multi-way interaction. Importantly, these rank-one components additively reconstruct the tensor to form a multi-linear model and do not have to depend on each other. This allows us to identify which entities significantly influence the interactions in each component, making it more straightforward to map those entities back to the original data and discover hidden patterns (Ho et al., 2014; Park et al., 2016; Al-Sayouri et al., 2020). For example,

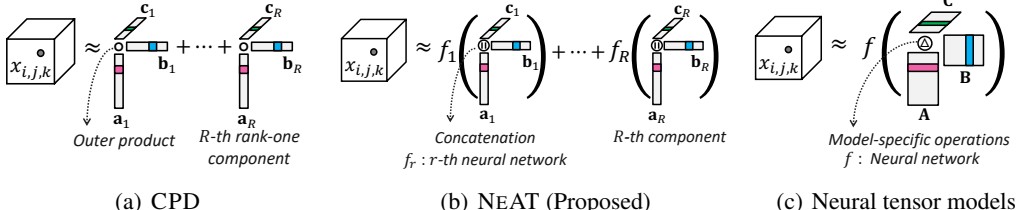

Figure 1: Comparison of model designs of CPD, NEAT, and neural tensor models with the third mode tensor. NEAT models $r$-th components separately to others with each neural network to achieve the interpretability from the rank-one tensor in CPD while neural tensor models entangle all the interactions between components.

movie rating data, where users rate movies at a specific time, can be represented as a tensor with modes (movie, user, time) with rating values. We can easily discover a soft clusterings of movies, users, and points in time by analyzing rank-one components obtained from CPD.

Even though CPD is preferred for interpretability to extract multi-linear structures, many real-world tensors are better explained via complex non-linear structures, which are often inadequately represented by addition of rank-one tensors. This limitation can lead to performance degradation in various practical applications. Recent tensor models for sparse tensors attempted to capture non-linear patterns based on neural networks, have attracted attention (He et al., 2017; Liu et al., 2018; 2019; Wu et al., 2019; Chen & Li, 2020; Tillinghast et al., 2020; Fan, 2021; Qian et al., 2022). Even though these approaches enhance the conventional tensor models with the expressive power of neural networks, the way they are designed to use neural networks intermixes components with each other thereby hindering the interpretability of latent components, as illustrated in Figure 1(c).

To address these challenges, we propose NEURAL ADDITIVE TENSOR DECOMPOSITION (NEAT) that accurately models non-linear latent structures, while also preserving properties of classical linear model CPD, which allows us to easily interpret the latent components. As illustrated in Figure 1(b), NEAT learns non-linear structures present in each component by employing individually parameterized neural networks, unlike existing neural tensor models that employ neural networks interacting with all components. Also, NEAT designs the model leveraging the sparsity of tensors to save significant computations by avoiding conventional tensor operations. Our contributions are summarized as follows:

- **Model.** We propose a novel neural tensor model NEAT, that can accurately learn non-linear structures while maintaining interpretability for sparse tensors.
- **Performance.** Extensive experiments on six real-world datasets demonstrate that NEAT shows the state-of-the-art performance in sparse tensor completion over multi-linear and neural tensor models and shows its ability to capture meaningful patterns in factors with downstream tasks.
- **Interpretability.** Finally we show how NEAT can serve as a glass-box model instead of a black box model and provide meaningful insights into its latent factors.

The rest of this paper is organized as follows. We introduce the preliminaries and related works in Sec. 2. We propose NEAT in Sec. 3 and present experimental results in Sec. 4. We summarize the key points and results of our paper in Sec. 5. The source code and datasets used in this paper are available at `https://anonymous.4open.science/r/NEAT`.

## 2 PRELIMINARIES & RELATED WORK

**Tensors** are defined as multi-dimensional arrays that generalize one-dimensional arrays (or vectors) and two-dimensional arrays (or matrices) to higher dimensions. Sparse tensors indicate tensors where majority of their entries are missing. Traditionally, the dimension of a tensor is referred to as its order or the number of modes; the size of each mode is called "dimensionality". We use boldface Euler script letters (e.g., $\mathcal{X}$) to denote tensors, boldface capitals (e.g., $\mathbf{A}$) to denote matrices, boldface lower cases (e.g., $\mathbf{a}$) to denote vectors. We denote the $i$-th row vector as $\mathbf{a}_{i,:}$, $r$-th column vector as $\mathbf{a}_r$ and $i, r$-th entry as $a_{ir}$.

**Canonical polyadic decomposition (CPD)** (Carroll & Chang, 1970; Harshman et al., 1970) approximates an $N$-th order tensor $\mathcal{X} \in \mathbb{R}^{I_1 \times \cdots \times I_N}$ as the sum of $R$ rank-one components as:

$$\mathcal{X} \approx [\![\mathbf{A}^{(1)}, \ldots, \mathbf{A}^{(N)}]\!] = \sum_{r=1}^{R} \mathbf{a}_1^{(1)} \circ \cdots \circ \mathbf{a}_R^{(N)} \tag{1}$$

where $\circ$ denotes an outer product, the $n$th factor matrix $\mathbf{A}^{(n)}$ represents entities in the $n$th mode, and $\mathbf{a}_r^{(n)} \in \mathbb{R}^{I_n}$ is the $r$th column vector of $\mathbf{A}^{(n)}$. The $r$th rank-one component $\mathbf{a}_1^{(1)} \circ \cdots \circ \mathbf{a}_R^{(N)}$ corresponds to the $r$th latent pattern which simultaneously clusters entities across $N$ modes. Also, Equation (1) is written as

$$x_\alpha \approx a_{i_1 1}^{(1)} a_{i_2 1}^{(2)} \ldots a_{i_N 1}^{(N)} + \cdots + a_{i_1 R}^{(1)} a_{i_2 R}^{(2)} \ldots a_{i_N R}^{(N)} = \sum_{r=1}^{R} \prod_{n=1}^{N} a_{i_n r}^{(n)} \tag{2}$$

where $x_\alpha$ indicates the $\alpha = (i_1, \cdots, i_N)$th entry of $\mathcal{X}$ and $a_{i_n r}^{(n)}$ indicates $(i_n, r)$-th element of $\mathbf{A}^{(n)}$, respectively. CPD reconstructs a given entry of a tensor with a sum of products of $r$th components of each row vector $\mathbf{a}_{i_n,:}^{(n)}$, which is also known as an embedding for $i_n$th entity in the $n$th mode.

**Additive Models.** CPD can be thought of as a multi-linear extension of Generalized additive models (GAMs) (Hastie & Tibshirani, 1990) where each additive component is the product of embedding values for that respective latent factor. More recently, works like Neural additive models (NAM) (Agarwal et al., 2021) extend GAMs to capture non-linear behavior in an additive manner. The formulation of NEAT shown in Equation (3) makes it a multi-modal extension of NAMs.

**Interpretability in Tensor Models.** In the realm of tensor decomposition, interpretability involves discovering hidden patterns in latent components related to original data more readily while interpretability generally refers to the ability to explain decisions made by a model in machine learning (Molnar, 2020). To discover hidden patterns with CPD, we identify the most influential entities in each component and treat them as a soft-clustering of the original tensor data (co-cluster). To further facilitate the interpretation of latent components, various methods make factor matrices non-negative and sparse (Ho et al., 2014; Afshar et al., 2021).

**Neural Tensor Models.** Traditionally, CPD and Tucker are good at fitting low-rank linear structures while they often fails to fit tensors including non-linear latent structures (Liu et al., 2019). Thus, numerous methods replace multi-linear operations with neural networks to capture complex structures. NCF (He et al., 2017) is a matrix factorization model that employs a Multi-layer perceptron (MLP) to learn multi-linear and non-linear interactions between users and items. NEURALCP (Liu et al., 2018) is a Bayesian tensor decomposition learning MLPs where its input is a long concatenation of row factors. NTF (Liu et al., 2018) exploits a Long short-term memory (LSTM) network for temporal interactions and MLP to model non-linear interactions between components for predictive tasks in dynamic relational data. COSTCO (Liu et al., 2019) leverages two Convolutional neural networks (CNNs) to capture nonlinear interaction across modes and ranks and uses MLPs for aggregating the output of CNNs. It claimed and showed that MLP-based tensor models are prone to overfit to sparse tensors due to its dense connections while CNNs avoids this problem. NTM (Chen & Li, 2020) combines two neural networks to learn multi-linear and non-linear relations considering the inner and outer products for recommendation tasks. POND (Tillinghast et al., 2020) is a probabilistic tensor decomposition leveraging Gaussian processes for capturing complex interactions and CNN to complete a given entry. $\text{M}^2\text{DMTF}$ (Fan, 2021) is a multi-mode nonlinear deep tensor factorization where each factor matrix is modeled with two-mode non-linear deep matrix factorization for tensor completion. JULIA (Qian et al., 2022) is a framework for a tensor decomposition model to jointly capture linear and non-linear interactions in the tensors by combining multi-linear and neural tensor models. However, the way these models rely on the interaction of different latent dimensions, makes it complicated to identify co-clusters in those dimensions. Additionally, interactions between components are more likely to learn spurious correlations between them, where components are uncorrelated but these false correlations are trained to fit tensors. In contrast, NEAT not only simplifies the discovery of patterns in its latent dimensions as CPD does but is also less likely to learn spurious correlations between components because each component does not rely on others. Furthermore, we show that, via NEAT, even using MLPs produces the best generalization if an appropriate design and training approach is used in contrast to this study Liu et al. (2019).

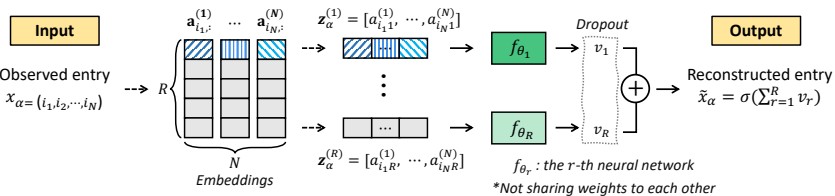

Figure 2: Illustration of a model architecture of NEAT for link prediction. NEAT reconstructs observed entries by summing outputs of individually parameterized networks, jointly capturing various patterns in factors and learning complex interactions between them with neural networks.

## 3 PROPOSED METHOD

We propose NEURAL ADDITIVE TENSOR DECOMPOSITION (NEAT) to accurately learn non-linear structures present in sparse tensors and to make those easy to interpret. We describe the details of the model in the following sections.

### 3.1 MODEL

A key challenge in designing our model arises from the question: How can we efficiently neuralize each component? In other words, how can we efficiently apply neural networks to each component? Given an $N$-mode tensor with the size $I_1 \times \cdots \times I_N$, processing all entries of components with neural networks equals to dealing with computations associated to the size of the given tensor. This is especially impractical for real-world sparse tensors that have high dimensionality, as presented in Table 1. To address this, NEAT leverages the sparsity of the tensor, considering only observed entries instead of the entire tensor. This approach is scalable to large tensors since it is irrelevant of tensor size and more accurate than the one of treating unobserved entries as 0 (Acar et al., 2011; Shin et al., 2016). As illustrated in Figure 2, NEAT gathers $N$ embeddings of dimension $R$ corresponding to an index $\alpha$, given a tensor entry $x_{\alpha=(i_1,\cdots,i_N)}$. Then $r$th neural network uses only $r$th entries of embeddings and captures non-linear interactions between them. NEAT reconstructs the tensor entry by summing all outputs from neural networks. Thus, neural networks operate on an input size $N$ for all observed entries rather than the entire entries, which significantly reduces computational cost.

Given an $N$th order tensor $\mathcal{X} \in \mathbb{R}^{I_1 \times \cdots \times I_N}$ with the observed entries and a rank $R$, NEAT aims to learn a set of factor matrices $\{\mathbf{A}^{(n)} \in \mathbb{R}^{I_n \times R} \mid 1 \leq n \leq N\}$ and a set of neural networks $\{f_{\theta_r} \mid 1 \leq r \leq R\}$, by reconstructing observed tensor entries $x_\alpha (\forall \alpha \in \Omega)$ as follows:

$$x_\alpha \approx f_{\theta_1}(\mathbf{z}_\alpha^{(1)}) + \cdots f_{\theta_r}(\mathbf{z}_\alpha^{(r)}) + \cdots f_{\theta_R}(\mathbf{z}_\alpha^{(R)}) \tag{3}$$

where $\Omega$ denotes the set of the observed indices, $\mathbf{z}_\alpha^{(r)} = \left[ a_{i_1 r}^{(1)}, a_{i_2 r}^{(2)}, \ldots, a_{i_N r}^{(N)} \right] \in \mathbb{R}^{1 \times N}$ is a concatenation of the $r$th element $a_{i_n r}^{(n)}$ of embeddings corresponding to the $\alpha$, and $f_{\theta_r}$ indicates the $r$th neural network. Each individual neural network operates on $r$th latent components $\mathbf{z}_\alpha^{(r)}$, returns $r$th contribution to a given entry $x_\alpha$, and their all outputs are then aggregated to reconstruct to $x_\alpha$.

It is important to note that each $f_{\theta_r}$ is individually parameterized and does not share parameters with each other. This allows each component act as an individual model, capturing distinct patterns and interactions and thereby accurately reconstructing a tensor. Interestingly, this design choice naturally introduces an ensemble-like characteristics, which capture different aspects of latent patterns in data and mitigate the impact of individual model errors, such that the collective prediction performance of ensembles is superior to that of a single model (Sagi & Rokach, 2018).

We employ MLPs to capture non-linear interactions between $r$-th components, defined as:

$$f_{\theta_r}(\mathbf{z}_\alpha^{(r)}) = \mathbf{h}_r^{(l-1)} \mathbf{W}_r^{(L)} + b_r^{(L)} \tag{4}$$

where $\mathbf{W}_r^{(L)} \in \mathbb{R}^{d_{L-1} \times 1}$ and $\mathbf{b}_r^{(L)} \in \mathbb{R}^{1 \times 1}$ are a weight and bias of the last $L$th layer. In the first layer, $\mathbf{h}_r^{(1)} \in \mathbb{R}^{1 \times d_1}$ is computed as:

$$\mathbf{h}_r^{(1)} = g(\mathbf{z}_\alpha^{(r)} \mathbf{W}_r^{(0)} + \mathbf{b}_r^{(0)}) \tag{5}$$

where $\mathbf{W}_r^{(0)} \in \mathbb{R}^{N \times d_1}$ and $\mathbf{b}_r^{(0)} \in \mathbb{R}^{1 \times d_1}$. For the subsequent layers ($l \geq 2$), we have:

$$\mathbf{h}_r^{(l-1)} = g(\mathbf{h}_r^{(l-2)} \mathbf{W}_r^{(l-1)} + \mathbf{b}_r^{(l-1)}) \tag{6}$$

where $\mathbf{W}_r^{(l-1)} \in \mathbb{R}^{d_{l-1} \times d_l}$ and $\mathbb{R}^{1 \times d_l}$. Note that $g$, $l$ ($1 \leq l \leq L$), and $d_l$ denote an activation function, a depth of layers, and the dimension size of a layer, respectively. We use the Rectified Linear Unit (ReLU) as an activation function for each internal layer, except for the last layer.

In Section 4, we show that NEAT with 2-layer MLPs excels compared to all the baselines; however we note that other types of neural networks can possibly replace the MLPs to further improve the accuracy, and we leave it as a future work. Further, if we replace $f_{\theta_r}$ as the function that returns the product of all entries in a vector, NEAT in Equation (3) is exactly the same as CPD in Equation (2), therefore making NEAT a generalizable extension of CPD. NEAT is able to express non-linear interactions by employing neural networks while CPD is able to only capture multi-linear interactions by computing an outer product between factor matrices.

## 3.2 TRAINING

In our experiments, we primarily focus on the completion of binary tensors, also known as a link prediction, and as a result, we opt for a binary cross-entropy loss function:

$$\mathcal{L}(\Theta) = -\frac{1}{|\Omega|} \sum_{\forall \alpha \in \Omega} (x_\alpha \log \tilde{x}_\alpha + (1 - x_\alpha) \log(1 - \tilde{x}_\alpha)) + \lambda R(\Theta) \tag{7}$$

where $x_\alpha$ and $\tilde{x}_\alpha$ indicates a observed entry and reconstruction corresponding to $\alpha$, respectively. We apply the Sigmoid function to a reconstructed entry $\tilde{x}_\alpha$ to predict the probability of observed entries for the link prediction task. $\lambda R(\Theta)$ is a regularization term for all parameters followed as:

$$R(\Theta) = \sum_{n=1}^{N} \|\mathbf{A}^{(n)}\|_F^2 + \sum_{r=1}^{R} \sum_{\mathbf{W}_r^{(l)} \in \theta_r} \|\mathbf{W}_r^{(l)}\|_F^2 \tag{8}$$

where $\lambda$ indicates a weight decay, $\mathbf{A}^{(n)}$ indicates the $n$th factor matrix, and $\mathbf{W}_r^{(l)}$ indicates $l$th weights in each $r$th neural network. In the case of real-valued tensors, we would accordingly modify our loss such as least squares, however, for simplicity of demonstrating the main point of the proposed method we defer to future work.

We optimize Equation (7) of NEAT based on Adam (Kingma & Ba, 2014) using backpropagation and jointly train factor matrices and MLPs. Although in principle a joint model like ours should be easier to train with backpropagation, we observe that only specific components are in utilization for loss minimization while others are not, resulting in poor performance. To avoid this, we apply Dropout (Srivastava et al., 2014) to all final outputs $f_{\theta_r}(\mathbf{z}_\alpha^{(r)})$ of MLPs. Dropout helps components to be trained appropriately by randomly selecting different subsets of components to reconstruct a tensor. Also, Figure 5(b) in Section 4.4 exhibits that applying Dropout is highly effective in better generalization. We further normalize inputs for MLPs and apply Dropout to each layer inside neural networks, excluding the final one for further stable training.

## 3.3 COMPLEXITY ANALYSIS

We analyze space and time complexities of our model and summarize the complexities of all baselines in Table 3 in Appendix A.1. The total dimensionality of an $N$-mode tensor is denoted as $I = I_1 + \cdots I_N$, where $I_n$ represents the size of the $n$th mode, and the rank size is denoted as $R$. The depth of a neural network and maximum size of a dimension in neural networks is denoted as $L$ and $D$ where $max(d_1, \cdots, d_l) = D$, respectively. NEAT has $IR$ number of parameters for factor matrices and approximately $RLD^2$ for neural networks such that each $r$th neural network has approximately equal to $LD^2$ parameters ($Nd_1 + d_1 d_2 + \cdots + d_{l-1} d_l + d_l$). Consequently, the total number of parameters in our model can be $\mathcal{O}(IR + RLD^2)$. The time complexity of NEAT is linear with regard to the number of observed entries and irrelevant to the size of a given tensor. When forwarding a single tensor entry through each neural network, it takes $RLD^2$ computational cost. When considering all observed entries, the overall computational complexity is $\mathcal{O}(|\Omega| RLD^2)$.

As shown in Section 4.4, we employ shallow networks (e.g., two layers) for most of experiments, showing the best link prediction performance. With the two-layer MLPs, computational costs for processing each entry becomes much smaller such as $\mathcal{O}(RD)$. Also, the parameters of neural networks are much less than the factors $\mathcal{O}(IR)$ since $I$ is much larger than $D$.

Table 1: Summary of six real-world sparse tensors.

| Name | Dimensionality | Nonzeros | Name | Dimensionality | Nonzeros |
|---|---|---|---|---|---|
| **DBLP** | $4{,}057 \times 14{,}328 \times 7{,}723$ | 94,022 | **MovieLens** | $610 \times 9{,}724 \times 4{,}110$ | 100,836 |
| **FS-NYC** | $1{,}084 \times 38{,}334 \times 7{,}641$ | 225,701 | **YELP** | $70{,}818 \times 15{,}580 \times 109$ | 335,022 |
| **FS-TKY** | $2{,}294 \times 61{,}859 \times 7{,}641$ | 570,743 | **Yahoo-M** | $82{,}309 \times 82{,}308 \times 168$ | 785,749 |

## 4 EXPERIMENTS

We conduct experiments to answer the following questions.

**Q1 Performance (Section 4.2).** How accurately does NEAT perform in link prediction?
**Q2 Pattern Discovery (Section 4.3).** Can NEAT learn meaningful patterns in components?
**Q3 Hyper-parameter Study (Section 4.4).** How do hyper-parameter settings affect performance?

### 4.1 EXPERIMENTAL SETTING

We conduct experiments on a machine equipped with an AMD Ryzen CPU and an NVIDIA RTX A6000 and describe the experimental setup in the following paragraphs.

**Datasets.** We use six real-world sparse tensors to evaluate the performance of the proposed method to the baselines. The datasets are summarized in Table 1 and details are provided in Table 4 in Appendix A.2. **MovieLens** (Harper & Konstan, 2015), **YELP**, and **Yahoo-M** are movie, business, and music rating datasets consisting of (user, item, timestamp). **FS-NYC** (Yang et al., 2015) and **FS-TKY** (Yang et al., 2015) are check-in datasets consisting of (user, venue, timestamp), collected by Foursquare in New York and Tokyo, respectively. Each entry $x_{i,j,k}$ of each tensor is binary indicating if a user $i$ is associated with an item $j$ (e.g., movie, venue) at the timestamp $k$. **DBLP** is a computer science bibliography network consisting of (author, paper, terminology) , representing whether an author $i$ published a paper $j$ including a terminology $k$. We split a tensor into training, validation, and test datasets with an 8:1:1 ratio. We randomly sample negative samples with the same number of observed entries as in the split dataset.

**Baselines.** We compare NEAT to five baselines which consist of multi-linear and neural tensor decomposition methods. CPD is a standard tensor model with an L2 regularization optimized by gradient descent. TUCKER (Balažević et al., 2019)[1] is a Tucker decomposition method for knowledge graph completion. NCF (He et al., 2017)[2] is a neural collaborative filtering model extended for tensors. COSTCO (Liu et al., 2019)[3] is a tensor completion model learning non-linear interactions with two 1-d CNNs and 2-layer MLPs. NTM (Chen & Li, 2020) is a tensor decomposition model that combines the inner product and the neuralized outer product via neural networks.

**Training.** We employ Adam to optimize all models and train all baselines except for CPD with a binary cross entropy as specified in Equation (7). We select all hyper-parameters via a combination of grid search and bayesian optimization based on early stopping. We find learning rates from $\{10^{-2}, 10^{-3}, 10^{-4}\}$, weight decays from $\{10^{-3}, 10^{-4}, 10^{-5}\}$, and ranks from $\{8, 16, 32, 64, 128\}$ for all models. We also find dimension sizes of layers from $\{8, 16, 32, 64, 128\}$, and layer depths from two to four, and batch sizes from $\{512, 1024\}$ for all neural tensor models.

### 4.2 LINK PREDICTION

We evaluate NEAT and baselines on the link prediction task in terms of accuracy with six real-world sparse tensors across different rank or embedding sizes. For all models, we repeat experimental results averaged over three runs with optimal hyper-parameters and report the total number of trained parameters used in Table 5 in Appendix A.3. We further evaluate the performance of NEAT and baselines on the link prediction with extremely sparse tensors in Tables 6 to 8 in Appendix A.4. Table 2 describes that NEAT consistently demonstrates superior accuracy compared to all baselines

---

[1] https://github.com/ibalazevic/TuckER
[2] https://github.com/guoyang9/NCF
[3] https://github.com/USC-Melady/KDD19-CoSTCo

Table 2: Accuracy of NEAT and baselines in link prediction. NEAT is superior to all baselines in six real-world sparse tensors across different rank sizes. Note that the best-performing method: bold, multi-linear tensor model: ♡, neural tensor model: ◇, and additive tensor model: ♣.

| | DBLP | | | | | MovieLens | | | | | YELP | | | | |
|---|---|---|---|---|---|---|---|---|---|---|---|---|---|---|---|
| Model \ Rank | 8 | 16 | 32 | 64 | 128 | 8 | 16 | 32 | 64 | 128 | 8 | 16 | 32 | 64 | 128 |
| CPD♡,♣ | 0.918 | 0.926 | 0.925 | 0.917 | 0.903 | 0.911 | 0.918 | 0.923 | 0.923 | 0.916 | 0.781 | 0.772 | 0.765 | 0.761 | 0.760 |
| TUCKER♡ | 0.834 | 0.837 | 0.946 | 0.966 | 0.965 | 0.902 | 0.919 | 0.934 | 0.941 | 0.944 | 0.829 | 0.831 | 0.831 | 0.833 | 0.826 |
| NCF◇ | 0.834 | 0.836 | 0.821 | 0.937 | 0.879 | **0.981** | **0.985** | 0.987 | 0.989 | 0.988 | **0.840** | **0.846** | 0.849 | 0.849 | 0.852 |
| COSTCO◇ | 0.933 | 0.948 | 0.956 | 0.939 | 0.932 | 0.978 | 0.983 | 0.986 | 0.989 | 0.990 | 0.838 | 0.846 | 0.849 | **0.857** | 0.858 |
| NTM◇ | 0.885 | 0.911 | 0.887 | 0.882 | 0.828 | 0.953 | 0.948 | 0.954 | 0.958 | 0.959 | 0.796 | 0.798 | 0.807 | 0.827 | 0.830 |
| NEAT◇,♣ | **0.942** | **0.960** | **0.969** | **0.973** | **0.976** | **0.981** | 0.984 | **0.988** | **0.990** | **0.991** | 0.835 | 0.836 | **0.854** | **0.857** | **0.864** |

| | FS-NYC | | | | | FS-TKY | | | | | Yahoo-M | | | | |
|---|---|---|---|---|---|---|---|---|---|---|---|---|---|---|---|
| Model \ Rank | 8 | 16 | 32 | 64 | 128 | 8 | 16 | 32 | 64 | 128 | 8 | 16 | 32 | 64 | 128 |
| CPD♡,♣ | 0.803 | 0.816 | 0.824 | 0.821 | 0.787 | **0.869** | **0.876** | 0.870 | 0.863 | 0.809 | 0.802 | 0.794 | 0.820 | 0.811 | 0.804 |
| TUCKER♡ | 0.805 | 0.811 | 0.816 | 0.829 | 0.838 | 0.852 | 0.854 | 0.870 | 0.876 | 0.881 | 0.866 | 0.879 | 0.886 | 0.891 | 0.908 |
| NCF◇ | 0.794 | 0.796 | 0.817 | 0.818 | 0.825 | 0.854 | 0.860 | 0.859 | 0.861 | 0.875 | 0.826 | 0.851 | 0.845 | 0.870 | 0.888 |
| COSTCO◇ | 0.806 | 0.816 | 0.824 | 0.833 | 0.833 | 0.862 | 0.869 | 0.875 | 0.876 | 0.878 | 0.841 | 0.806 | 0.819 | 0.845 | 0.838 |
| NTM◇ | 0.762 | 0.776 | 0.778 | 0.796 | 0.805 | 0.833 | 0.834 | 0.839 | 0.843 | 0.844 | 0.836 | 0.833 | 0.828 | 0.846 | 0.857 |
| NEAT◇,♣ | **0.811** | **0.830** | **0.845** | **0.851** | **0.849** | 0.861 | 0.873 | **0.881** | **0.887** | **0.887** | **0.917** | **0.927** | **0.925** | **0.918** | **0.915** |

including multi-linear and neural tensor models across all six datasets. Especially, NEAT trained with rank eight shows the biggest performance gap between competitors for **Yahoo-M**, the most sparse and largest tensor we used; it achieves 5% and 7% points higher accuracy than the second-best multi-linear and neural tensor model, TUCKER and COSTCO trained with the same rank, respectively. Also, NEAT consistently shows a stable and increasing trend in performance when the rank size increases. This indicates that individually parameterized components capture more diverse hidden structures by using more components. The performance of multi-linear models, CPD and TUCKER, improves as the rank increases. However, the performance of CPD declines at larger ranks (e.g., 64 or 128); it tends to overfit the data when the rank increases, capturing noises rather than meaningful multi-linear patterns. Also, TUCKER requires a significant amount of parameters ($R^N$) due to a core tensor when the rank size is higher, still, it does not achieve the best performance. For **MovieLens** and **YELP**, neural tensor models, NCF, COSTCO, and NTM, show better accuracy over multi-linear tensor models even at lower ranks (e.g., 8 or 16) and NEAT achieves the best accuracy with higher ranks. This indicates that there exist various complex latent patterns in those two datasets, thus conventional multi-linear models fail to fit complex tensors. For **DBLP** and **FS-TKY**, CPD performs well at lower ranks, indicating that these two datasets include linear patterns dominantly. Thus, neural tensor models trained with even higher ranks show marginal improvements in performance over multi-linear models. However, NEAT achieves better performance for all rank sizes, which means that NeAT captures latent structures accurately even if the tensor includes little non-linear patterns. For **FS-NYC** and **Yahoo-M**, neural tensor models and multi-linear tensor models show competitive performance to each other. These two tensors are highly sparse among all the datasets we had, which indicates their little information seems to be noisy and makes it difficult for tensor models to capture non-linear or linear patterns. Also, neural tensor models may capture spurious correlations between latent dimensions to accurately fit sparse tensors. Especially, COSTCO generally performs better than NCF and NTM due to CNNs, which learn informative patterns using convolutional filters, which are much smaller than heavily parameterized MLPs. According to the study (Liu et al., 2019), MLP-based neural tensor models are prone to overfitting sparse tensors due to MLP's excessive over-parameterization in the form of redundant connections. NEAT avoids the issue of overparameterization of learning spurious correlations between latent dimensions, via additive components with shallow MLPs, and in doing so achieves the best generalization performance over all baselines for sparse tensors.

## 4.3 PATTERN DISCOVERY

We conduct an experiment to show if NEAT is able to capture meaningful patterns in components in Section 4.3.1 and also visualize discovered patterns from learned factors in Section 4.3.2. Thanks to simplicity of NEAT, we are able to easily visualize and analyze patterns.

### 4.3.1 DOWNSTREAM TASK

We evaluate NEAT and baselines on a downstream task using **DBLP** dataset consisting of (author, paper, conference). The task is to classify the research area of authors with author embeddings obtained from tensor models. There are two settings: a transductive and inductive setting. Under the transductive setting, an input tensor includes both training and test author samples. We simultaneously obtain training and test embeddings from tensor models. In the inductive setting, we train tensor models with only a training tensor, consisting of training authors, and inference test embeddings for new authors with the trained models. To produce test embeddings, we freeze all parameters in tensor models except for test author embeddings. We train test embeddings with frozen tensor models by reconstructing a test tensor until they converge. Note that the parameters of multi linear and neural tensor models indicate factor matrices and a core tensor, and the factor matrix and all weights of neural networks, respectively. To focus on evaluating the embeddings themselves, we employ a linear classifier rather than advanced classifiers.

Figure 3(a) exhibits stable classification performance with embeddings obtained from NEAT in transductive setting, indicating that the embeddings are able to capture meaningful patterns. Figure 3(b) shows that NEAT and neural tensor models better classification performance than multi-linear tensor models in inductive setting. This indicates that trained neural networks accurately learn complex interactions between latent patterns. Therefore, NEAT and neural tensor models are able to capture both meaningful patterns and complex interactions by jointly training neural networks and factor matrices. They are advantageous for both transductive and inductive settings.

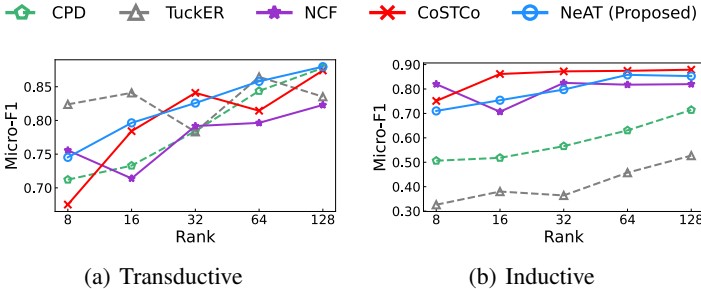

(a) Transductive  (b) Inductive

Figure 3: Comparison of NEAT and baselines in terms of Micro-F1 for the downstream task. NEAT presents better generalization compared to multi-linear models for both inductive and transductive settings since it captures meaningful patterns in factors and complex interactions with MLPs.

### 4.3.2 VISUALIZATION

We visualize discovered latent patterns from factor matrices obtained from NEAT with a rank of eight. Thanks to simplicity of NEAT, we are able to easily explore factor matrices to discover patterns by considering only each column (component) of factor matrices without interactions between components. To explore latent patterns, we consider the top-$k$ highest valued factors in each factor matrix. We use Inverse document frequency (IDF) scores (Park et al., 2016) to consider top-$k$ entities that appear in only a few columns. Each author and conference entity is labeled with one of four areas of study: Information Retrieval (IR), Database (DB), Data Mining (DM), and Artificial Intelligence (AI). Figure 4 present coherent pattern discoveries. For example, Figures 4(a) and 4(b) reveal that the second and fourth components are softly clustered based on DB and IR labels, respectively. Figure 4(c) highlights that DB and IR have higher values in each component. Further, by examining the top-$k$ authors in each component, we observe that each component is homogeneous with respect to its main label. We display the entire conference factor matrix and label distribution for all components are in Figures 9 and 10.

### 4.4 HYPER-PARAMETERS STUDY

We study major hyper-parameters in NEAT: rank, dimensions of layers, and Dropout ratio. We explore the effect of the number of trainable parameters for the performance with regard to rank and the dimension size of neural networks. Note that the white color in the heatmap indicates the blank

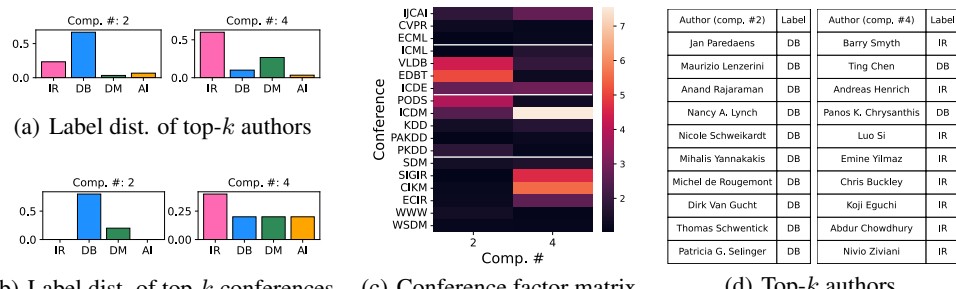

Figure 4: Pattern discoveries obtained from NEAT exhibit coherent results. (a) and (b) indicates the label distributions of top-$k$ authors and conferences. (c) indicates the second and fourth columns of the conference factor matrix where its y-axis is sorted in the order of AI, DB, DM, and IR labels. (d) highlights the top-$k$ authors in the second and the fourth components.

since bayesian optimization did not visit those hyper-parameter combinations. We increase the rank and dimension size ranging from 8, 16, 32, 64, 128 and use the 2-layer MLPs, and evaluate link prediction accuracy for all datasets as shown in Figure 5(a). Also, Figure 8 indicates that using shallow MLPs generally shows the best performance than using deeper networks. Note that increasing the rank size and dimension size improves accuracy for all datasets except for `FS-TKY`. This indicates that NEAT is able to capture various patterns in factors while learning complex interactions between them with even shallow MLPs. Additionally, we investigate the impact of Dropout ratio $p$ on the final outputs for the performance. We vary the dropout ratio ranging from 0 to 0.9 and evaluate accuracy in Figure 5(b). The sensitivity of NEAT on rank and Dropout ratio shows similar patterns for all datasets; applying Dropout provides significant improvements even at lower ranks; higher Dropout ratio is more suitable for the larger ranks. This indicates that Dropout can lead to better generalization by giving a chance to all components be trained evenly.

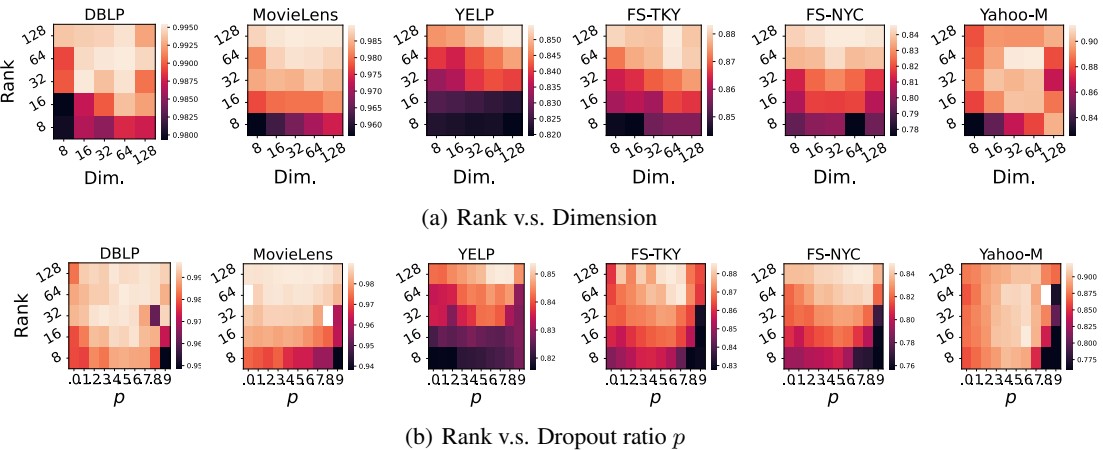

Figure 5: Effect of rank, dimension size, and dropout in NEAT. NEAT improves performance with larger ranks and dimension, and higher Dropout ratio.

## 5 CONCLUSION

We propose a neural tensor model that neuralizes each latent component in an additive fashion, which captures various patterns and complex interactions in real-world sparse tensors and lends themselves to direct and intuitive interpretations. Experiment results demonstrate the effectiveness of our proposed method for tensor analysis and can be applied to a downstream task. One limitation of the paper is lack of theoretical analysis in effectiveness of additive components over non-additive components. Interestingly, the design and training approach of NEAT resembles an ensemble learning. In the future work, we will analyze the both generalization and optimization of additive components with a connection to ensemble learning.

## REPRODUCIBILITY

We recognize the critical importance of reproducibility in advancing scientific knowledge. To enhance the replicability of our work, we have taken several measures. We provide datasets and the implementation of the proposed method in anonymously downloadable source code `https://anonymous.4open.science/r/NEAT`. We also provide the link to downloadable source code for baselines and the link to download the original datasets we used. Furthermore, to facilitate the replication of our experiments, we provide a comprehensive description of the data processing steps for the used datasets and hyper-parameter settings in the paper. We believe that these efforts contribute to the transparency and reliability of our research.

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
