# A APPENDIX

## A.1 COMPLEXITY ANALYSIS

Table 3: Number of parameters for each model considered. We denote mode as $N$, rank as $R$, all dimensionality of mode as $I$, a layer depth as $L$, and the maximum and minimum hidden dimension size of neural networks as $D$ and $d$, respectively. Note that we consider the number of parameters of NCF for tensors.

| Model | # of params |
|---|---|
| CPD | $IR$ |
| NCF (He et al., 2017) | $2IR + (R + D)D + LD^2$ |
| CoSTCo (Liu et al., 2019) | $IR + RD + ND^2 + D^2$ |
| TUCKER (Balažević et al., 2019) | $IR + R^N$ |
| NTM (Chen & Li, 2020) | $IR + LND^2 + d^3$ |
| NEAT (Proposed) | $IR + LRD^2$ |

## A.2 DATASET

Table 4 summarizes statistics of real-world sparse tensors. We calculate the size and sparsity of tensors by taking the product of all dimensionalities (e.g., $I_1 \times \cdots \times I_N$) and dividing their nonzeros by their size, respectively. In our datasets, **MovieLens** exhibits the smallest size and is less sparse than other datasets, while **FS-TKY** and **Yahoo-M** exhibit the largest size and are more sparse than other datasets.

Table 4: Statistics of real-world sparse tensors.

| Dataset | Dimensionality | # Nonzeros | Size | Sparsity |
|---|---|---|---|---|
| **DBLP** [1] | $4,057 \times 14,328 \times 7,723$ | 94,022 | 4.5e+11 | 2.1e-07 |
| **MovieLens** [2] | $610 \times 9,724 \times 4,110$ | 100,836 | 2.4e+10 | 4.1e-06 |
| **YELP** [3] | $70,818 \times 15,580 \times 109$ | 335,022 | 1.2e+11 | 2.8e-06 |
| **FS-NYC** [4] | $1,084 \times 38,334 \times 7,641$ | 225,701 | 3.2e+11 | 7.1e-07 |
| **FS-TKY** [4] | $2,294 \times 61,859 \times 7,641$ | 570,743 | 1.1e+12 | 5.3e-07 |
| **Yahoo-M** [5] | $82,309 \times 82,308 \times 168$ | 785,749 | 1.1e+12 | 6.9e-07 |

[1] https://github.com/Jhy1993/HAN
[2] https://files.grouplens.org/datasets/movielens/ml-latest-small-README.html
[3] https://www.yelp.com/dataset
[4] https://sites.google.com/site/yangdingqi/home/foursquare-dataset
[5] https://webscope.sandbox.yahoo.com/catalog.php?datatype=r

## A.3 PARAMETERS

We report the number of trainable parameters of all models evaluated on link prediction performance corresponding to Table 2. Tables 2 and 5 describe that NEAT achieves superior performance with relatively smaller parameters. Interestingly, NEAT requires more parameters at lower ranks for **MovieLens** and **YELP** to capture complex interactions in tensors. Note that NCF generally includes the biggest parameters since it consists of two sets of factor matrices since the sum of all dimensionalities $I$ is much larger than the dimension sizes of neural networks as mentioned earlier in Section 3.3. CoSTCo and TUCKER are the models includes the second-biggest parameters. Due to the core tensor, TUCKER includes biggest parameters at higher ranks.

## A.4 SPARSITY

We further evaluate NEAT with baselines by making **MovieLens**, **FS-TKY**, and **Yahoo-M** datasets extremely sparse. With a sampling ratio of 10%, 30%, and 50%, we simulate the extreme sparsity with only training tensors and evaluate models on the test tensors. We select the two best neural tensor models NCF and CoSTCo and the other two multi-linear models as baselines. Tables 6 to 8 demonstrates that even with the extreme sparsity, NEAT shows superior performance

Table 5: Comparison of the number of trainable parameters of NEAT and baselines for link prediction across different rank size. Note that the best-performing and second-best method denoted in bold and underline.

| Model \ Rank | DBLP 8 | 16 | 32 | 64 | 128 | MovieLens 8 | 16 | 32 | 64 | 128 | YELP 8 | 16 | 32 | 64 | 128 |
|---|---|---|---|---|---|---|---|---|---|---|---|---|---|---|---|
| CPD♡,♣ | 2.1e+05 | 4.2e+05 | 8.4e+05 | 1.7e+06 | 3.3e+06 | 1.2e+05 | 2.3e+05 | 4.6e+05 | 9.2e+05 | 1.8e+06 | 6.9e+05 | 1.4e+06 | 2.8e+06 | 5.5e+06 | 1.1e+07 |
| TUCKER♡ | 2.1e+05 | 4.2e+05 | 8.7e+05 | 1.9e+06 | 5.4e+06 | 1.2e+05 | 2.4e+05 | 4.9e+05 | 1.2e+06 | 3.9e+06 | 6.9e+05 | 1.4e+06 | 2.8e+06 | 5.8e+06 | 1.3e+07 |
| NCF◇ | **4.2e+05** | **8.4e+05** | **1.7e+06** | **3.4e+06** | **6.7e+06** | 2.3e+05 | 4.7e+05 | 9.3e+05 | 1.9e+06 | 3.7e+06 | 1.4e+06 | 2.8e+06 | 5.5e+06 | 1.1e+07 | 2.2e+07 |
| CoSTCo◇ | 3.6e+05 | 7e+05 | 9.7e+05 | 2.7e+06 | 3.9e+06 | 1.2e+05 | 2.3e+05 | 4.7e+05 | 9.9e+05 | **4e+06** | 6.9e+05 | 1.7e+06 | 3.3e+06 | 6.6e+06 | 1.3e+07 |
| NTM◇ | 2.2e+05 | 4.3e+05 | 8.4e+05 | 1.7e+06 | 3.4e+06 | 1.2e+05 | 2.3e+05 | 4.6e+05 | 9.3e+05 | 1.9e+06 | 6.9e+05 | 1.4e+06 | 2.8e+06 | 5.5e+06 | 1.1e+07 |
| NEAT◇,♣ | 2.1e+05 | 4.2e+05 | 8.4e+05 | 1.7e+06 | 3.4e+06 | **2.5e+05** | 2.7e+05 | 4.6e+05 | 9.4e+05 | 1.9e+06 | 7.6e+05 | 1.7e+06 | 2.8e+06 | 5.6e+06 | 1.1e+07 |

| Model \ Rank | FS-NYC 8 | 16 | 32 | 64 | 128 | FS-TKY 8 | 16 | 32 | 64 | 128 | Yahoo-M 8 | 16 | 32 | 64 | 128 |
|---|---|---|---|---|---|---|---|---|---|---|---|---|---|---|---|
| CPD♡,♣ | 3.8e+05 | 7.5e+05 | 1.5e+06 | 3e+06 | 6e+06 | 5.7e+05 | 1.1e+06 | 2.3e+06 | 4.6e+06 | 9.2e+06 | 1.3e+06 | 2.6e+06 | 5.3e+06 | 1.1e+07 | 2.1e+07 |
| TUCKER♡ | 3.8e+05 | 7.6e+05 | 1.5e+06 | 3.3e+06 | 8.1e+06 | 5.7e+05 | 1.2e+06 | 2.3e+06 | 4.9e+06 | 1.1e+07 | 1.3e+06 | 2.6e+06 | 5.3e+06 | 1.1e+07 | 2.3e+07 |
| NCF◇ | **7.5e+05** | **1.5e+06** | **3e+06** | **6e+06** | **1.2e+07** | **1.2e+06** | **2.3e+06** | **4.6e+06** | **9.2e+06** | **1.8e+07** | **2.6e+06** | **5.3e+06** | **1.1e+07** | **2.1e+07** | **4.2e+07** |
| CoSTCo◇ | 5.2e+05 | 1e+06 | 2e+06 | 4.1e+06 | 8.1e+06 | 7.2e+05 | 1.4e+06 | 2.8e+06 | 5.7e+06 | 1.1e+07 | 1.3e+06 | 2.6e+06 | 5.3e+06 | 1.1e+07 | 2.3e+07 |
| NTM◇ | 3.8e+05 | 7.5e+05 | 1.5e+06 | 3e+06 | 6e+06 | 5.8e+05 | 1.2e+06 | 2.3e+06 | 4.6e+06 | 9.2e+06 | 1.3e+06 | 2.6e+06 | 5.3e+06 | 1.1e+07 | 2.1e+07 |
| NEAT◇,♣ | 3.8e+05 | 7.5e+05 | 1.5e+06 | 3.1e+06 | 6.1e+06 | 5.8e+05 | 1.2e+06 | 2.3e+06 | 4.6e+06 | 9.2e+06 | 1.3e+06 | 2.6e+06 | 5.3e+06 | 1.1e+07 | 2.1e+07 |

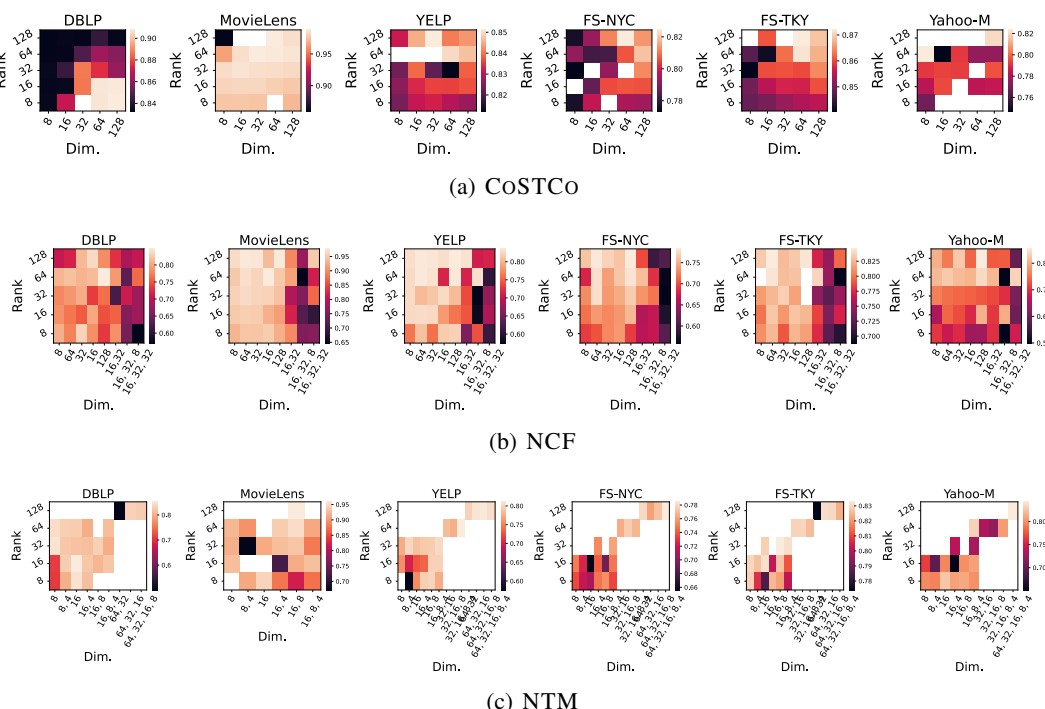

(a) CoSTCo

(b) NCF

(c) NTM

Figure 6: Accuracy of neural tensor models in terms of rank and dimension size. Note that white color indicates the blank since bayesian optimization did not visit those combination of hyperparameters but chose the best combinations.

for **FS-TKY** and **Yahoo-M**. Especially for **Yahoo-M**, there is a significant accuracy gap between NEAT and baselines. The performance of all baselines trained with 10% of **Yahoo-M** is poor, indicating that they underfit to data. When 10% of **MovieLens** is sampled, NCF and CoSTCo shows better performances than other models. This indicates that interactions between components are helpful if the complex tensor is extremely sparse. However, NEAT may potentially improve the performance if we replace the simple MLPs with more complicated neural networks (e.g., CNNs). One of the advantages of NEAT is that we can flexibly change the neural networks due to its simple yet effective design and optimization technique.

Table 6: Comparison of link prediction accuracy of NEAT compared to baselines with different sparsity sampling ratios for **MovieLens**.

| Sparsity | 10% | | | | | 30% | | | | | 50% | | | | |
|---|---|---|---|---|---|---|---|---|---|---|---|---|---|---|---|
| Model \ Rank | 8 | 16 | 32 | 64 | 128 | 8 | 16 | 32 | 64 | 128 | 8 | 16 | 32 | 64 | 128 |
| CPD$^{\heartsuit,\clubsuit}$ | 0.743 | 0.742 | 0.739 | 0.737 | 0.730 | 0.803 | 0.809 | 0.812 | 0.807 | 0.794 | 0.83 | 0.837 | 0.843 | 0.840 | 0.824 |
| TUCKER$^{\heartsuit}$ | 0.499 | 0.500 | 0.499 | 0.499 | 0.499 | 0.705 | 0.769 | 0.809 | 0.822 | 0.834 | 0.832 | 0.837 | 0.850 | 0.880 | 0.891 |
| NCF$^{\diamondsuit}$ | **0.914** | 0.819 | 0.906 | **0.911** | **0.904** | 0.945 | **0.967** | **0.972** | 0.961 | 0.959 | **0.972** | **0.976** | **0.980** | 0.981 | 0.952 |
| CoSTCo$^{\diamondsuit}$ | 0.878 | **0.89** | **0.908** | 0.860 | 0.832 | **0.947** | 0.953 | 0.963 | 0.942 | 0.937 | 0.961 | 0.970 | 0.976 | 0.980 | 0.982 |
| NEAT$^{\diamondsuit,\clubsuit}$ | 0.857 | 0.871 | 0.867 | 0.875 | 0.873 | 0.940 | 0.950 | 0.961 | **0.969** | **0.971** | 0.962 | 0.970 | 0.979 | **0.984** | **0.985** |

Table 7: Comparison of link prediction accuracy in NEAT and baselines with different sparsity ratios with **FS-TKY**.

| Sparsity | 10% | | | | | 30% | | | | | 50% | | | | |
|---|---|---|---|---|---|---|---|---|---|---|---|---|---|---|---|
| Model \ Rank | 8 | 16 | 32 | 64 | 128 | 8 | 16 | 32 | 64 | 128 | 8 | 16 | 32 | 64 | 128 |
| CPD$^{\heartsuit,\clubsuit}$ | 0.721 | 0.709 | 0.701 | 0.694 | 0.693 | 0.762 | 0.761 | 0.753 | 0.749 | 0.742 | 0.783 | 0.786 | 0.782 | 0.772 | 0.764 |
| TUCKER$^{\heartsuit}$ | 0.498 | 0.498 | **0.785** | 0.497 | **0.788** | 0.622 | 0.500 | 0.826 | 0.831 | 0.827 | 0.829 | 0.834 | 0.836 | 0.837 | 0.835 |
| NCF$^{\diamondsuit}$ | 0.771 | 0.774 | 0.771 | 0.772 | 0.768 | 0.805 | 0.812 | 0.814 | 0.818 | 0.819 | 0.817 | 0.830 | 0.836 | 0.839 | 0.845 |
| CoSTCo$^{\diamondsuit}$ | 0.784 | 0.781 | 0.784 | **0.784** | 0.786 | **0.817** | **0.815** | 0.813 | 0.814 | 0.809 | 0.833 | 0.826 | 0.838 | 0.834 | 0.825 |
| NEAT$^{\diamondsuit,\clubsuit}$ | **0.786** | **0.789** | **0.785** | **0.784** | 0.784 | 0.810 | 0.811 | **0.832** | **0.842** | **0.844** | **0.837** | **0.850** | **0.857** | **0.860** | **0.864** |

Table 8: Comparison of link prediction accuracy in NEAT and baselines with different sparsity ratios with **Yahoo-M**.

| Sparsity | 10% | | | | | 30% | | | | | 50% | | | | |
|---|---|---|---|---|---|---|---|---|---|---|---|---|---|---|---|
| Model \ Rank | 8 | 16 | 32 | 64 | 128 | 8 | 16 | 32 | 64 | 128 | 8 | 16 | 32 | 64 | 128 |
| CPD$^{\heartsuit,\clubsuit}$ | 0.564 | 0.564 | 0.563 | 0.564 | 0.568 | 0.605 | 0.605 | 0.607 | 0.605 | 0.607 | 0.627 | 0.624 | 0.627 | 0.626 | 0.629 |
| TUCKER$^{\heartsuit}$ | 0.500 | 0.499 | 0.500 | 0.498 | 0.499 | 0.499 | 0.500 | 0.499 | 0.500 | 0.500 | 0.767 | 0.795 | 0.812 | 0.816 | 0.820 |
| NCF$^{\diamondsuit}$ | 0.502 | 0.501 | 0.502 | 0.501 | 0.504 | 0.711 | 0.715 | 0.715 | 0.719 | 0.727 | 0.704 | 0.747 | 0.747 | 0.757 | 0.781 |
| CoSTCo$^{\diamondsuit}$ | 0.511 | 0.511 | 0.511 | 0.511 | 0.500 | 0.560 | 0.531 | 0.500 | 0.515 | 0.515 | 0.534 | 0.535 | 0.535 | 0.516 | 0.500 |
| NEAT$^{\diamondsuit,\clubsuit}$ | **0.667** | **0.669** | **0.668** | **0.669** | **0.67** | **0.732** | **0.729** | **0.742** | **0.747** | **0.753** | **0.818** | **0.825** | **0.827** | **0.825** | **0.821** |

## A.5 CONVERGENCE & TRAINING TIME

We evaluate NEAT and models in terms of their averaged running time per epoch and convergence. NEAT shows a stable convergence and similar speed as other baselines.

## A.6 EXTRA HYPER-PARAMETER STUDY

We investigate the impact of layer depth on the performance of NEAT. We increase the depth from two to four and evaluate NEAT in Figure 8. Interestingly, two-layer MLPs of NEAT show consistently the best performance for all five datasets, indicating that shallow networks are able to effectively capture complex interactions.

## A.7 INTERPRETABILITY

The IDF score is defined as follows.

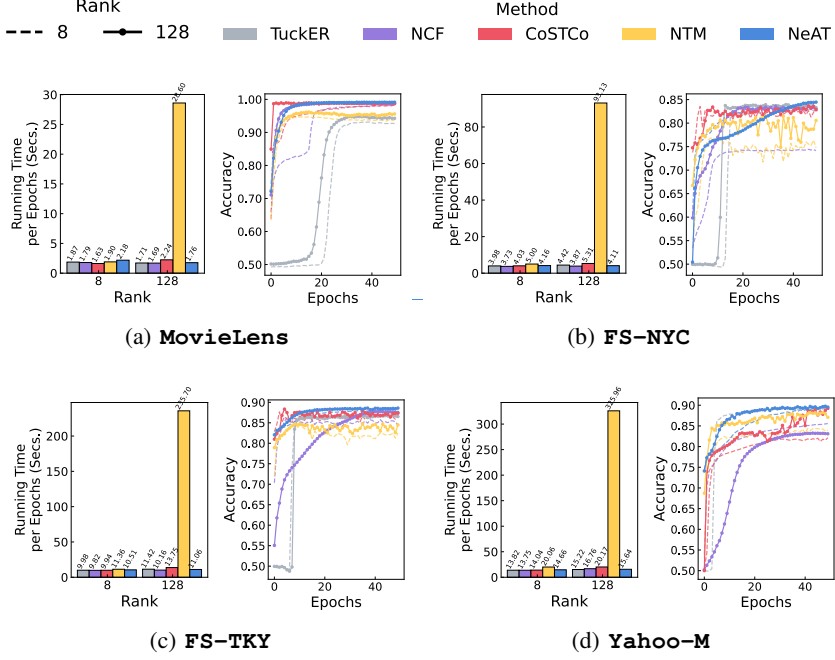

(a) **MovieLens**

(b) **FS-NYC**

(c) **FS-TKY**

(d) **Yahoo-M**

Figure 7: Comparison of NEAT and baselines in terms of convergence and training time in link prediction with the rank sizes 8 and 128.

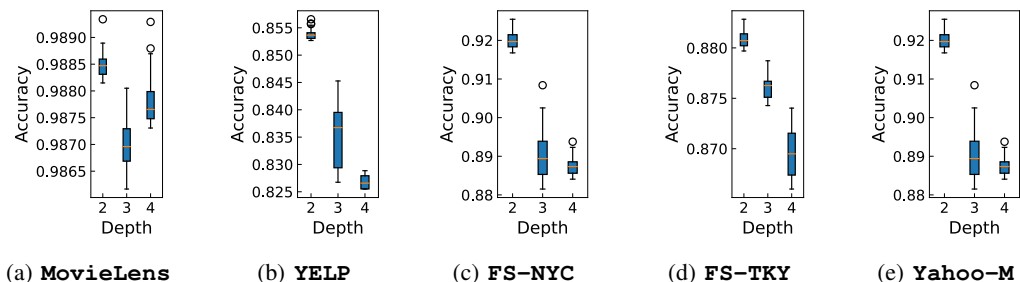

(a) **MovieLens**

(b) **YELP**

(c) **FS-NYC**

(d) **FS-TKY**

(e) **Yahoo-M**

Figure 8: NEAT performs well with two-layers MLPs.

$$IDF score = (1 + \alpha \log(R/col)) \times val \qquad (9)$$

where $\alpha$ is a constant, $R$ is the rank, $col$ is the number of components where the entity appears in, and $val$ is the value of factors.

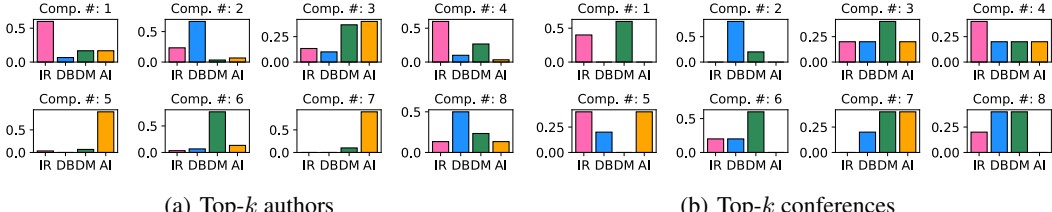

(a) Top-$k$ authors          (b) Top-$k$ conferences

Figure 9: Label distribution of top-$k$ authors and conferences.

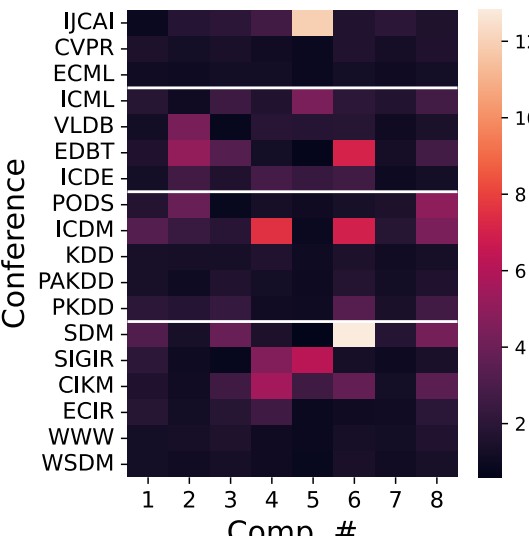

Figure 10: Conference factor matrix.

## A.8 EXTRA RESULTS

Table 9: A performance comparison of NEAT with baselines in link prediction. NEAT is superior to all baselines in six real-world datasets in three metrics. Note that the best-performing method: bold, $\heartsuit$: multi-linear tensor model, $\diamondsuit$: neural tensor model, and $\clubsuit$: additive tensor model.

| | DBLP | | | MovieLens | | | YELP | | |
|---|---|---|---|---|---|---|---|---|---|
| | Acc. | F1 | AUC | Acc. | F1 | AUC | Acc. | F1 | AUC |
| CPD$^{\heartsuit,\clubsuit}$ | $0.92_{\pm 0.002}$ | $0.71_{\pm 0.001}$ | $0.96_{\pm 0.001}$ | $0.92_{\pm 0.001}$ | $0.72_{\pm 0.000}$ | $0.96_{\pm 0.000}$ | $0.76_{\pm 0.000}$ | $0.67_{\pm 0.000}$ | $0.88_{\pm 0.001}$ |
| TUCKER$^{\heartsuit}$ | $0.95_{\pm 0.004}$ | $0.95_{\pm 0.003}$ | $\mathbf{0.99}_{\pm 0.002}$ | $0.93_{\pm 0.004}$ | $0.93_{\pm 0.005}$ | $0.98_{\pm 0.003}$ | $0.83_{\pm 0.002}$ | $0.83_{\pm 0.003}$ | $0.90_{\pm 0.002}$ |
| NCF$^{\diamondsuit}$ | $0.82_{\pm 0.143}$ | $0.77_{\pm 0.230}$ | $0.86_{\pm 0.173}$ | $\mathbf{0.99}_{\pm 0.001}$ | $\mathbf{0.99}_{\pm 0.001}$ | $\mathbf{1.00}_{\pm 0.000}$ | $\mathbf{0.85}_{\pm 0.001}$ | $\mathbf{0.85}_{\pm 0.001}$ | $\mathbf{0.92}_{\pm 0.002}$ |
| CoSTCo$^{\diamondsuit}$ | $0.96_{\pm 0.005}$ | $0.96_{\pm 0.004}$ | $\mathbf{0.99}_{\pm 0.000}$ | $\mathbf{0.99}_{\pm 0.001}$ | $\mathbf{0.99}_{\pm 0.001}$ | $\mathbf{1.00}_{\pm 0.000}$ | $\mathbf{0.85}_{\pm 0.004}$ | $\mathbf{0.85}_{\pm 0.002}$ | $\mathbf{0.92}_{\pm 0.001}$ |
| NTM$^{\diamondsuit}$ | $0.89_{\pm 0.001}$ | $0.89_{\pm 0.001}$ | $0.95_{\pm 0.002}$ | $0.95_{\pm 0.003}$ | $0.95_{\pm 0.004}$ | $0.99_{\pm 0.001}$ | $0.81_{\pm 0.001}$ | $0.79_{\pm 0.000}$ | $0.87_{\pm 0.001}$ |
| NEAT$^{\diamondsuit,\clubsuit}$ | $\mathbf{0.97}_{\pm 0.001}$ | $\mathbf{0.97}_{\pm 0.001}$ | $\mathbf{0.99}_{\pm 0.001}$ | $\mathbf{0.99}_{\pm 0.001}$ | $\mathbf{0.99}_{\pm 0.001}$ | $\mathbf{1.00}_{\pm 0.000}$ | $\mathbf{0.85}_{\pm 0.000}$ | $\mathbf{0.85}_{\pm 0.000}$ | $\mathbf{0.92}_{\pm 0.001}$ |

| | FS-NYC | | | FS-TKY | | | Yahoo-M | | |
|---|---|---|---|---|---|---|---|---|---|
| | Acc. | F1 | AUC | Acc. | F1 | AUC | Acc. | F1 | AUC |
| CPD$^{\heartsuit,\clubsuit}$ | $0.82_{\pm 0.001}$ | $0.67_{\pm 0.000}$ | $0.86_{\pm 0.000}$ | $0.87_{\pm 0.000}$ | $0.69_{\pm 0.001}$ | $0.90_{\pm 0.001}$ | $0.82_{\pm 0.004}$ | $0.70_{\pm 0.001}$ | $0.88_{\pm 0.002}$ |
| TUCKER$^{\heartsuit}$ | $0.82_{\pm 0.001}$ | $0.81_{\pm 0.001}$ | $0.88_{\pm 0.001}$ | $0.87_{\pm 0.001}$ | $\mathbf{0.87}_{\pm 0.001}$ | $0.92_{\pm 0.001}$ | $0.89_{\pm 0.001}$ | $0.89_{\pm 0.001}$ | $0.94_{\pm 0.000}$ |
| NCF$^{\diamondsuit}$ | $0.82_{\pm 0.004}$ | $0.82_{\pm 0.003}$ | $0.88_{\pm 0.002}$ | $0.83_{\pm 0.029}$ | $0.81_{\pm 0.049}$ | $0.87_{\pm 0.060}$ | $0.84_{\pm 0.008}$ | $0.84_{\pm 0.004}$ | $0.88_{\pm 0.009}$ |
| CoSTCo$^{\diamondsuit}$ | $0.82_{\pm 0.008}$ | $0.82_{\pm 0.005}$ | $0.89_{\pm 0.001}$ | $\mathbf{0.88}_{\pm 0.001}$ | $\mathbf{0.87}_{\pm 0.001}$ | $\mathbf{0.93}_{\pm 0.001}$ | $0.82_{\pm 0.006}$ | $0.82_{\pm 0.006}$ | $0.88_{\pm 0.005}$ |
| NTM$^{\diamondsuit}$ | $0.78_{\pm 0.001}$ | $0.77_{\pm 0.003}$ | $0.84_{\pm 0.002}$ | $0.84_{\pm 0.001}$ | $0.83_{\pm 0.001}$ | $0.90_{\pm 0.001}$ | $0.83_{\pm 0.001}$ | $0.83_{\pm 0.001}$ | $0.89_{\pm 0.001}$ |
| NEAT$^{\diamondsuit,\clubsuit}$ | $\mathbf{0.84}_{\pm 0.003}$ | $\mathbf{0.83}_{\pm 0.002}$ | $\mathbf{0.90}_{\pm 0.001}$ | $\mathbf{0.88}_{\pm 0.001}$ | $\mathbf{0.87}_{\pm 0.001}$ | $\mathbf{0.93}_{\pm 0.001}$ | $\mathbf{0.92}_{\pm 0.001}$ | $\mathbf{0.92}_{\pm 0.001}$ | $\mathbf{0.97}_{\pm 0.001}$ |

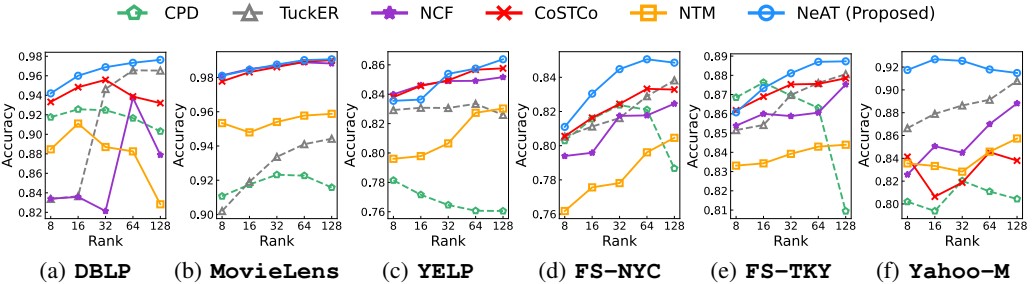

(a) DBLP  (b) MovieLens  (c) YELP  (d) FS-NYC  (e) FS-TKY  (f) Yahoo-M

Figure 11: Comparison of NEAT and baselines in terms Accuracy in link prediction across varying rank sizes. NEAT has a consistent performance across the rank size.

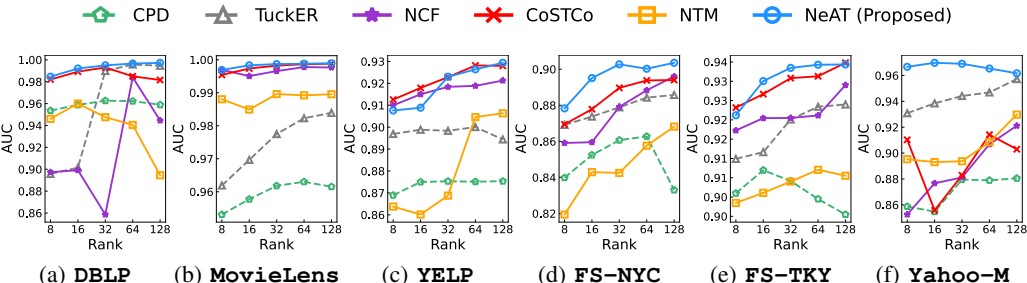

(a) DBLP  (b) MovieLens  (c) YELP  (d) FS-NYC  (e) FS-TKY  (f) Yahoo-M

Figure 12: Comparison of NEAT and baselines in terms of AUC in link prediction across varying rank sizes. NEAT has a consistent performance cross the rank size.