# OpenReview forum: "NEURAL ADDITIVE TENSOR DECOMPOSITION FOR SPARSE TENSORS"
_ICLR.cc/2024/Conference — Submitted to ICLR 2024_

### Official Review · Reviewer_LDaU · 2023-10-31

**Soundness:** 4 excellent
**Presentation:** 3 good
**Contribution:** 2 fair
**Rating:** 3
**Confidence:** 4

**Summary:**

The paper considers a modified CP decomposition in which the multi-vector inner product defining each element is replaced by a sum of neural networks, whose input dimension is equal to the order of the tensor. The experiments focus on tensor completion for binary tensors, with a cross-entropy loss function. The experiments include comparison with alternate decompositions for accuracy on a few tensor completion test problems. Additionally performance and interpretability is considered for classification tasks.

**Strengths:**

+ The proposed model is relatively simple and makes sense, though theoretical/application grounding of the method is not quite clear.
 + The results demonstrate improvements in accuracy over a variety of baselines, which seem to be achieved with a more compact model (though model size is not explicitly evaluated).
 + The results include both performance evaluation and downstream tasks.
 + The paper is clear / reasonably well-written, in both presentation of the method and its evaluation.

**Weaknesses:**

- Besides the definition of the new algorithm, the paper has practically no theoretical analysis, besides a simple cost quantification based on CP parameters.
 - In discussion of closely related works on tensor decompositions with factors replaced by neural nets, the following sentence is used as contrast "However, the way they used neural network entangled associations of all components makes it difficult to identify the contribution of entities." This seems overly broad, and should be justified by particular aspects of each of the related methods.
 - Comparison of different models in accuracy for relative to model size would be of interest.
 - The accuracy improvements compared to other models seem pretty minor.
 - I have some concerns regarding whether state-of-the-art methods are being used as baselines here, please see the questions.

Overall, the paper pursues an interesting hybrid method that combines a neural network with tensor decomposition, and shows that it is competitive for tensor completion. However, there is little theoretical motivation or analysis to motivate the effectiveness of the approach. The experimental results point to improvements in accuracy, but make some simplifying assumptions on baseline methods that make it difficult to draw strong conclusions from the results. Given that ICLR has a theoretical focus, I would recommend rejection of the paper in its current form, but I believe the ideas and experimental studies are of value and should be published after revision.

**Questions:**

* Are the baselines implemented by the authors or are existing codes used? I would be concerned that Adam is not an efficient method (does not lead to the more accurate decomposition) for training some of the baselines.
 * Why is CPD not trained relative to cross-entropy? There are existing works for CP completion with generalized loss functions.

---

> ### Author Response · Authors · 2023-11-23
> **Responses to Weaknesses**
>
> > Besides the definition of the new algorithm, the paper has practically no theoretical analysis, besides a simple cost quantification based on CP parameters.
> * In lieu of making simplifying theoretical  assumptions for analytically studying nonlinear models like  NeAT, we evaluate the proposed method in four different ways, which in our view makes it a more empirically justifiable model. We also improved our analysis of empirical evaluations in a revised version.
>     * Generalization performance (Section 4.2, Appendix 4): We evaluate NeAT with five baselines in link prediction in terms of accuracy with six real-world datasets by varying rank as sizes shown in Table 2. We further evaluate all models in the scenario where the tensors are extremely sparse in Tables 6 to 8. NeAT shows superior performance for all baselines for all datasets.
>     * The usefulness of components (Section 4.3): We evaluate if NeAT is able to capture meaningful patterns in components with downstream tasks and visualization. Note that we easily analyze patterns obtained from NeAT thanks to its simplicity. Those results show that factors include meaningful patterns and MLPs learn complex interactions as well.
>     * Training convergence and speed (Appendix 5): We evaluate NeAT with baselines in terms of convergence and training speed. NeAT shows that it converges stable and practically it is fast as other models.
>     * Sensitivity to hyper-parameters (Section 4.4 & Appendix 6):
>     We investigate important hyper-parameters of models to examine the performance of NeAT.
> * In case you find our empirical analysis limited, we would like to incorporate a more theoretically aligned analysis of NeAT along the lines you may suggest.
>
>
> > In discussion of closely related works on tensor decompositions with factors replaced by neural nets, the following sentence is used as contrast "However, the way they used neural network entangled associations of all components makes it difficult to identify the contribution of entities." This seems overly broad, and should be justified by particular aspects of each of the related methods.
> * We significantly changed the Neural Tensor Models paragraph which includes the corresponding sentence by highlighting the difference between neural tensor models and NeAT.
>
>
> > Comparison of different models in accuracy relative to model size would be of interest.
> * We would like to mention first that NeAT performs well with shallow MLPs, using smaller parameters than other neural tensor models and TuckER.
> * We have reported the number of parameters trained for all models in Table 5 where performance corresponds to Table 2.
> We also added Figure 6 in the Appendix 3. Note that the white color in the heatmap indicates the blank where Bayesian optimization does not explore to find the best combination hyper-parameters. Also in the original paper of NTM, they use the smaller dimension size of neural network than rank size.
>
> > The accuracy improvements compared to other models seem pretty minor
> * We significantly improved the analysis of experiment setting and we further added experimental results in terms of tensor sparsity.
> Note that we changed the main performance table by re-arranging old results in Table 2 and Figure 3 in the original draft into Table 2 in the revision. We move those into Appendix 8 temporarily for your convenience.
>     * We compare models by varying rank sizes instead of fixing the rank size since it is difficult to compare performance between models
>     * We report only Accuracy instead of three metrics since all metrics show a consistent performance trend.
>     * We change the floating points from two to three to round performance. The previous one has no space to show the values till three floating points.

---

> ### Author Response · Authors · 2023-11-23
> **Responses to Weaknesses**
>
> > I have some concerns regarding whether state-of-the-art methods are being used as baselines here, please see the questions.
> * Three neural tensor models, NCF, CoSTCo, and NTM, used for comparison, are important baselines to compare the performance as they are cited in many papers: CoSTCo[1-5], NCF[1, 5, 6], and NTM [1]
> * Even though we already mentioned a few more SOTA in related works, we have not compared them for the following reasons.
>     * JULIA [2]: this paper proposed the tensor framework which unifies existing multi-linear and neural tensor models while we propose a new neural tensor model. In some sense, this is complementary work that we may consider investigating in conjunction with NeAT as future work. Furthermore, the code of this model is proprietary and not publicly available.
>     * D^2DMTF [3]: this model is not designed for large-scale sparse tensors and is not scalable to the tensors used for evaluating models in the paper. Thus, we tried it with the official source code and had out-of-memory results for the model with the datasets we used.
>     * AGH [1]: this model does not provide a public implementation of the proposed algorithm and we could not obtain the source code
>
>
>
> > Overall, the paper pursues an interesting hybrid method that combines a neural network with tensor decomposition and shows that it is competitive for tensor completion.
> However, there is little theoretical motivation or analysis to motivate the effectiveness of the approach. The experimental results point to improvements in accuracy, but make some simplifying assumptions on baseline methods that make it difficult to draw strong conclusions from the results. Given that ICLR has a theoretical focus, I would recommend rejection of the paper in its current form, but I believe the ideas and experimental studies are of value and should be published after revision.
> * Thanks to the reviewer for acknowledging that the method is interesting.
> * Regarding the analysis of the effectiveness of the approach, we significantly improved the explanation and added to the revision;
> we have added a new explanation about empirical evaluation in Sections 4.2 to 4.4 and A.3 to A.7.
> We also further evaluate models with extremely sparse tensors in Tables 6 to 8, describing NeAT outperforms all baselines.
> * About theoretical motivation, our intent behind this paper is to introduce the NeAT model as a neural extension of the CPD tensor decomposition.
> As a result, the contributions in this first piece of work are empirical, albeit starting from a basis (the CPD model) which has been widely studied theoretically.
> In this paper, as we are focusing on demonstrating the efficacy and utility of the model, we focus on rigorously evaluating different aspects of it experimentally, and we defer theoretical analysis for future work.
> * We mentioned our possible future direction in Section 5.
>
>
> [1] Hui, Bo, and Wei-Shinn Ku. "Low-rank Nonnegative Tensor Decomposition in Hyperbolic Space." Proceedings of the 28th ACM SIGKDD Conference on Knowledge Discovery and Data Mining. 2022.
>
> [2] Qian, Cheng, et al. "Julia: Joint multi-linear and nonlinear identification for tensor completion." arXiv preprint arXiv:2202.00071 (2022).
>
> [3] Fan, Jicong. "Multi-mode deep matrix and tensor factorization." international conference on learning representations. 2021.
>
> [4] Oh, Sejoon, et al. "Influence-guided data augmentation for neural tensor completion." Proceedings of the 30th ACM International Conference on Information & Knowledge Management. 2021.
>
> [5] Chen, Huiyuan, and Jing Li. "Neural tensor model for learning multi-aspect factors in recommender systems." International Joint Conference on Artificial Intelligence (IJCAI). Vol. 2020.
>
> [6] Wu, Xian, et al. "Neural tensor factorization for temporal interaction learning." Proceedings of the Twelfth ACM international conference on web search and data mining. 2019.

---

> ### Author Response · Authors · 2023-11-23
> **Responses to Questions**
>
> > Are the baselines implemented by the authors or are existing codes used? I would be concerned that Adam is not an efficient method (does not lead to a more accurate decomposition) for training some of the baselines.
> * CoSTCo is available, but we implemented the pytorch version of the original one based on TensorFlow and we check whether the implementation produces the same results as the original one.
> We use the open source code for NCF which is able to replicate the performance of the original NCF and we modify the model for the tensor, which is a simple modification.
> We carefully implemented the code of NTF from scratch since the code is not available and we use the original code for TuckER.
> * I added the footnotes for all baseline links in section 4.1 baseline paragraph.
>
> > Why is CPD not trained relative to cross-entropy? There are existing works for CP completion with generalized loss functions.
> * CPD trained with cross-entropy performs worse than CPD with reconstruction loss in our setting.
> * Note that CPD-BCE indicates that CPD trained with Binary cross entropy loss and CPD-Rec indicates that CPD trained with a reconstruction loss.
>
>
> | Dataset        | TF      |     8 |    16 |    32 |    64 |   128 |
> |:---------------|:--------|------:|------:|------:|------:|------:|
> | DBLP           | CPD-Rec | **0.919** | **0.926** | **0.927** | **0.919** | **0.905** |
> |                | CPD-BCE | 0.856 | 0.851 | 0.859 | 0.719 | 0.638 |
> | MovieLens      | CPD-Rec | 0.911 | 0.918 | 0.924 | 0.924 | 0.917 |
> |                | CPD-BCE | **0.925** | **0.935** | **0.938** | **0.937** | **0.935** |
> | YELP           | CPD-Rec | 0.782 | 0.772 | **0.765** | **0.761** | 0.761 |
> |                | CPD-BCE | **0.786** | **0.794** | 0.657 | 0.646 | **0.793** |
> | FS-NYC         | CPD-Rec |**0.804** | **0.817** | **0.825** | **0.825** | **0.789** |
> |                | CPD-BCE | 0.758 | 0.758 | 0.762 | 0.754 | 0.686 |
> | FS-TKY         | CPD-Rec | **0.869** | **0.878** | **0.870** | **0.865** | **0.810** |
> |                | CPD-BCE | 0.825 | 0.823 | 0.808 | 0.809 | 0.805 |
> | Yahoo-M        | CPD-Rec | **0.804** | **0.796** | **0.823** | **0.814** | **0.806** |
> |                | CPD-BCE | 0.578 | 0.548 | 0.532 | 0.527 | 0.557 |

---

> ### Author Response · Authors · 2023-11-23
> **Thank you**
>
> Thank you for your constructive comments and valuable suggestions. We'd appreciate that reviewer also think the method is interesting and it is new algorithm. In the revision, as we emphasize that our paper focuses on emprical aspects of models, we significantly strengthened the analysis of the evaluation and presented it as clearly as possible. We hope that it will further help you understand the empirical results. Also, especially below two comments are highly useful for authors to think again about differences between competitors and NeAT in terms of generalization effects. We'd really appricate it again.
>
> > In discussion of closely related works on tensor decompositions with factors replaced by neural nets, the following sentence is used as contrast "However, the way they used neural network entangled associations of all components makes it difficult to identify the contribution of entities." This seems overly broad, and should be justified by particular aspects of each of the related methods.
>
> > Comparison of different models in accuracy for relative to model size would be of interest.

---

### Official Review · Reviewer_ZGpK · 2023-10-31

**Soundness:** 2 fair
**Presentation:** 3 good
**Contribution:** 2 fair
**Rating:** 5
**Confidence:** 4

**Summary:**

This paper proposes a neural tensor decomposition method called NEAT that captures nonlinear patterns in sparse tensors while maintaining interpretability. Aligned with classical CPD, NEAT decomposes a tensor into a sum of components, where each component is modeled by a separate MLP. This allows the expression of nonlinearities in an additive, separable way. NEAT is evaluated on link prediction for multiple real-world sparse tensors. NEAT also produces interpretable embeddings that allow the discovery of patterns in different components.

**Strengths:**

The proposed model shows a novel and reasonable way to combine the simple but interpretable CPD with the deep module. Besides state-of-the-art link prediction performance on multiple datasets, the design of the down-streaming task with finetuning is interesting. Also, the presentation is smooth and easy to follow.

**Weaknesses:**

- Compared to prior deep tensor work, the main selling point of the work is the interpretability based on the component(rank)-wise modeling. The experiment part (section 4.4) also highlights it. However, as component-wise nonlinear MLPs are used, do the learned latent factors really reflect the useful pattern in data? In other words, is it possible that meaningful patterns will be encoded in component-wise MLP, otherwise the latent factor's component?  More discussion or numerical experiments to investigate the consistency are encouraged.

- As I recognize the importance of the interpretability of tensor factor, is the interpretability from component-independent formulation better or more helpful than such component-cross methods? For example, for Tucker decomposition, the learned Tucker core can also show interpretability to help people understand the inner pattern of data. Some claims on why component-wise interpretability is crucial should be highlighted. Otherwise, the novelty and contribution of the proposed work could be limited.

- The experiment setting and analysis could be further enhanced. Some other non-linear tensor methods could be added as baselines, such as Gaussian-Process-based[1]. The downstream task setting by finetuning is interesting, but some discussion and analysis on why the proposed design could enhance the generalization performance are encouraged.

[1]Tillinghast, Conor, et al. "Probabilistic neural-kernel tensor decomposition." 2020 IEEE International Conference on Data Mining (ICDM). IEEE, 2020.

**Questions:**

See weakness

---

> ### Author Response · Authors · 2023-11-23
> **Responses to Weaknesses**
>
> > Compared to prior deep tensor work, the main selling point of the work is the interpretability based on the component(rank)-wise modeling. The experiment part (section 4.4) also highlights it. However, as component-wise nonlinear MLPs are used, do the learned latent factors really reflect the useful pattern in data? In other words, is it possible that meaningful patterns will be encoded in component-wise MLP, otherwise the latent factor's component? More discussion or numerical experiments to investigate the consistency are encouraged.
> * This is a good point. We have already evaluated this aspect to check if NeAT is able to capture useful patterns in components in Sections 4.4 and 4.5 for downstream tasks and Interpretability in the previous draft. However, we did not clearly articulate the purpose of the conducted experiments and visualization. Therefore, we rearrange Sections 4.4 and 4.5 (old) into Sections 4.3.1 and 4.3.2 (new). We clearly explain the purpose and the results as follows.
>     * Section 4.3: Pattern discovery
>         * Section 4.3.1 Downstream task
>         * Section 4.3.2 Visualization
> * According to Figure 3 in Section 4.3.1, NeAT shows stable performance when the rank increases in both transductive and inductive settings, which indicates that it captures meaningful patterns in factors rather than overfitting data when the rank increases. Also, it outperforms multi-linear models in the inductive setting, which indicates that NeAT also effectively learns complex interactions with MLPs.
> * According to Figure 4 in Section 4.3.2, we can identify that NeAT also produces coherent patterns in factors via visualizations.
>
>
> > As I recognize the importance of the interpretability of tensor factor, is the interpretability from component-independent formulation better or more helpful than such component-cross methods? For example, for Tucker decomposition, the learned Tucker core can also show interpretability to help people understand the inner pattern of data. Some claims on why component-wise interpretability is crucial should be highlighted. Otherwise, the novelty and contribution of the proposed work could be limited.
> * Even though we already explained how CPD-like interpretability is useful for interpretability in General response, we will explain briefly again.
> * With regard to simplicity, additive components from CPD and NeAT allow users to discover the hidden patterns easily rather than interlinked components (Tucker or other neural tensor models). When we analyze factors, we need to consider the interactions between components for accurate analysis, which complicates the process of discovering patterns.
> * We further discussed how the interpretability of CPD can be different from the Tucker model.
> Tucker (in an unconstrained form) returns the subspace bases for the rows, columns, and third-mode fibers of a tensor. The columns of those basis matrices are *not* latent factors, they just tell us how we can generate any row/col/fiber in the data as a linear combination of those columns. As a result, if we would like to use those Tucker factor matrices, we would have to either constrain them to *not* be orthogonal (in which case, we are deviating from the Tucker model) and we would probably have to constrain the core to be sparse. However, in this case, we can always rewrite that “Tucker-like” model as a CPD model, by repeating columns appropriately and eliminating the non-diagonal entries of the core tensor (and this would not necessarily hurt uniqueness, since all currently known uniqueness guarantees for CPD allow for column repetition in some of the factor matrices, since they require the sum of the k-ranks for the factor matrices to be bounded).

---

> ### Author Response · Authors · 2023-11-23
> **Responses to Weaknesses**
>
> > The experiment setting and analysis could be further enhanced.
> * We significantly improved the analysis of experimental evaluation and we further added experimental results in terms of tensor sparsity.
> * Note that we changed the main performance table by re-arranging old results in Table 2 and Figure 3 in the original draft into **Table 2 in the revision**. We move those into Appendix 8 temporarily for your convenience.
>     * We compare models by varying rank sizes instead of fixing the rank size since it is difficult to compare performance between models
>     * We report only Accuracy instead of three metrics since all metrics show a consistent performance trend.
>     * We change the floating points from two to three to round performance. The previous one has no space to show the values till three floating points.
>
> > Some other non-linear tensor methods could be added as baselines, such as Gaussian-Process-based[1]. The downstream task setting by finetuning is interesting, but some discussion and analysis on why the proposed design could enhance the generalization performance are encouraged.
> [1]Tillinghast, Conor, et al. "Probabilistic neural-kernel tensor decomposition." 2020 IEEE International Conference on Data Mining (ICDM). IEEE, 2020.
> * Also thanks to the reviewer, we omitted the suggested method in the previous draft, we included the study in the introduction and related works.
> * We tried to add this method as a baseline with the official source code but we obtained out-of-memory results on our machine when the rank size increased while the original paper indicates that it can increase the rank by 20.
> It will take time to further explore the model to work and finish full sets of experiments.
> * However, we will report small experiment results that we  were able to conduct as below.
>     * **Accuracy** of POND trained with rank 3 on **MovieLens** with different sampling ratios of 10%, 30%, and 50%.
>    * Note that POND trained with rank 3 and NeAT trained with rank 8 therefore it is not a fair comparison.
>
> | Sparsity | POND (3) | NeAT (8) |
> |---------:|---------:|---------:|
> |      0.5 |    0.951 |   **0.962** |
> |      0.3 |    0.929 |    **0.940** |
> |      0.1 |    **0.862** |    0.857 |

---

> ### Author Response · Authors · 2023-11-23
> **Thank you**
>
> Thank you for your constructive comments and valuable insights. We'd appreciate that reviewer also think the method is novel and our evaluation way is interesting. In the revision, we have tried to clarify why CPD-like interpretability is important and useful for real-world applications (also in general responses).  Furthermore, we strengthened the analysis of the evaluation and presented it clearly. We hope that it will help you understand the empirical results. Finally, thank you for suggesting the baseline for evaluating our model. Unfortunaley, we did not include the baseline at this stage as we were running out of rebuttal period. However, we are willing to add it after this rebuttal period (It is worth to add it since it outperforms CoSTCo).

---

> > ### Comment · Reviewer_ZGpK · 2023-11-23
> >
> > I appreciate the detailed rebuttal and updated draft from the authors, which did improve the paper. The highlight of the motivation and evaluation settings is more clear now. However, I decided to keep my current score, as I agree with some shared concerns about novelty from other reviewers.
> >
> > My other suggestion for authors is to post the response earlier, instead of waiting till the last day. It's too rushed for reviewers to check responses' details and post more discussions.

---

> ### Author Response · Authors · 2023-11-23
>
> We would appreciate your response and apologize for the delay in responses.
> Could you kindly specify which aspect of the novelty you concerned?
> We have thoroughly reviewed all the comments provided by the reviewers, and it appears that none explicitly mentioned novelty, apart from discussions around contribution, which we have addressed comprehensively in our General responses.
>
> Additionally, we carefully addressed reviewer ZGpK's concerns related to novelty in terms of interpretability and provided detailed explanations. If there are any aspects we may not address sufficiently, we would greatly value your insights.

---

> > ### Comment · Reviewer_ZGpK · 2023-11-23
> >
> > I read again on authors’ response on the novelty and interpretability, and decided to raise my score to 6. Still, please do the rebuttal earlier!

---

### Official Review · Reviewer_3Eaz · 2023-11-01

**Soundness:** 2 fair
**Presentation:** 2 fair
**Contribution:** 1 poor
**Rating:** 3
**Confidence:** 4

**Summary:**

This paper proposed a neural network model (NEAT) for sparse tensor decomposition. NEAT replaces the direct inner products of factors in CP decomposition with learnable MLPs. The authors showed NEAT outperformed CP and other neural network based tensor decomposition models in several real world tensor completion datasets and one downstream task. The authors showed the latern factors learned by NEAT can be used for interpretability.

**Strengths:**

The paper is clearly written and fairly easy to follow. The authors conduct thorough experiments to show the performance of the proposed model.

**Weaknesses:**

* The contribution is somewhat limited. Rank-k CP decomposition is the sum of the outer products of k rank-1 factors. The proposed NEAT model uses learnable MLPs to replace the outer product. There have been many papers doing similar things, some of which are cited in this paper as well. It is unclear how the proposed NEAT model is different comparing with the existing neural network based models. From section 2, it seems the biggest advantage of NEAT is that there's no cross factor interaction, as NEAT is a direct extension of CP decomposition model.
* The analysis of the empirical evaluation is not strong enough to justify the contribution. The chosen datasets are not complicated enough to demonstrate that NEAT significantly outperforms baselines. The interpretability part is good to have but is not good enough to convince readers that the proposed model is outstanding on that part.

**Questions:**

* In section 2, the authors mentioned that "... neural network entangled associations of all components makes it difficult to identify the contribution, ...  the proposed method NEAT ... simplifying the discovery of non-linear latent patterns".  Could the authors please elaborate on this? Does it always hold that simple methods are better? What types of tensors would be better fit by complex models, and what types of tensors would be better fit by the proposed model or CP model?
* How many extra parameters are introduced by NEAT when comparing with CP when the factors CP rank are the same? Could it be the case that, for some sparse tensors, the total number of parameters of NEAT is more than the number of observed entries? In that case, how does NEAT do in terms of overfitting?
* Does the proposed model work for non-sparse tensors?

---

> ### Author Response · Authors · 2023-11-22
> **Responses to Weaknesses**
>
> > The contribution is somewhat limited. Rank-k CP decomposition is the sum of the outer products of k rank-1 factors. The proposed NEAT model uses learnable MLPs to replace the outer product. There have been many papers doing similar things, some of which are cited in this paper as well.
> * There are three main contributions of the proposed model:  **Interpretability**, **Efficiency**, and **Accuracy**, ***which are not very well articulated in the previous draft while we carefully designed the model to achieve them***.
> We significantly added an explanation in the revisions but we will describe each one as follows reflecting the revision:
>
> * **Interpretability**
>     * The idea of neuralizing each component is the key for interpretability.
>     First, we are motivated by the CPD model, which can be useful for the completion or other downstream tasks,
>     but also in extracting a unique set of latent factors which represent the data and
>     lend themselves to the direct interpretation of their coefficients (as co-clustering coefficients, as mentioned in Sections 1 and 2).
>     As a result,  our intent is to “neuralize” each component that captures more complex latent structures while maintaining the interpretability of CPD.
>     * ***However, this concept can have two main challenges: efficiency and accuracy***. We address these problems as follows.
> * **Efficiency  (Section 3.1)**
>     * How can we neuralize each component efficiently? Each component has exactly the same size as the tensor size which is enormously huge to compute.
>     To address this, we employ the sparsity of sparse tensors to efficiently neuralize the component; We process each component by entry unit such that we can reduce the computational cost of neural networks.
>     The detailed explanation is presented in the first paragraph in Section 3.1.
>         * Note that exploiting sparsity for sparse tensors is a popular idea to model the design. However, we would like to emphasize that employing a sparsity of sparse tensors addresses the challenge that we faced to achieve the goal (neuralizing each component for interpretability).
>
> * **Accuracy (Sections 3.1 and 3.2)**
>     * According to this study [1], MLP-based tensor models are more likely to overfit sparse tensors due to MLP's dense connection.
>     However, we show that the MLP-based tensor model (NeAT) achieves the best performance if we carefully design the model and opt for a training approach.
>         * Model design: Each component (MLPs and factors) does not share parameters with each other. This can be seen as each component acts like an individual model. Thus, these components can be used to capture more diverse patterns in factors and learn complex interactions between them.
>         * Training: To optimize the loss of NeAT, we jointly train the model, MLPs and factors. If the complexity of MLPs is increasing, only specific components can be trained while others are not.
>         To prevent this, we add a dropout at all final outputs of neural networks, which indicates NeAT reconstructs tensor entries during training by randomly selecting subsets of components.
>             * Even though dropout is a common technique for deep learning models, we’d like to emphasize that we used a suitable regularization technique for the proposed model, considering the model design of NeAT.
>
> ***Finally, extensive sets of diverse experiment results consistently exhibit that NeAT shows superior performance in tensor completion tasks over multi-linear tensor and neural tensor models,
> due to its careful model design and appropriate training approach, even if tensors are extremely sparse (Tables 6 to 8).***
>
> [1] Hanpeng Liu, Yaguang Li, Michael Tsang, and Yan Liu. Costco: A neural tensor completion model for sparse tensors. In Proceedings of the 25th ACM SIGKDD International Conference on Knowledge Discovery & Data Mining, pp. 324–334, 2019.

---

> ### Author Response · Authors · 2023-11-22
> **Responses to Questions**
>
> > In section 2, the authors mentioned that "... neural network entangled associations of all components makes it difficult to identify the contribution, ... the proposed method NEAT ... simplifying the discovery of non-linear latent patterns". Could the authors please elaborate on this?
>
> * We apologize for the unclarity. We significantly changed the Neural Tensor Models paragraph which includes the corresponding sentence and highlighted the difference between neural tensor models and NeAT.
> * The original sentence means existing neural tensor models make all components interact with each other and this design makes interpretation of factors complicated.
>
> > Does it always hold that simple methods are better?
> * We already explained “why simplicity is important” in the previous responses (General response), but we will explain briefly again.
> * The simplicity of model design is directly related to tensor models’ interpretability since we find hidden patterns by examining the factors themselves.
> * Compared to Tucker, CPD examines only each column of factors while Tucker examines all interactions between components with a core tensor.
> Similar to CPD, NeAT makes all the components not interact with each other, making analyzing components easier. In other words, we do not need to analyze its components associated with others.
> * Also, once the model achieves the required performance, interpretability becomes more beneficial for real-world applications.
>
> > What types of tensors would be better fit by complex models, and what types of tensors would be better fit by the proposed model or CP model?
> * CPD is good at fitting tensors that have underlying low-dimensional linear subspaces that can easily be spanned by the column space of the factor matrices of CPD.
> If the underlying data manifold has any form of non-linearity, CPD fails to be a good fit, and neural tensor models offer better approximation capabilities that can fit those tensors, using neural networks.
> * According to experimental evaluation, NeAT and neural tensor models perform well on MovieLens and Yelp datasets over multi-linear models, and their performance increases when the rank size increases.
> This indicates that there exist various complex latent patterns in those two datasets, thus conventional multi-linear models fail to fit complex tensors.
> * For, FS-TKY and DBLP, CPD performs well at lower ranks, indicating that these two datasets include linear patterns dominantly.
> Thus, neural tensor models show marginal improvements even with higher ranks over multi-linear models.
> However NeAT achieves better performance for all rank sizes, which means that NeAT captures latent structures accurately even if the tensor includes little non-linear patterns.
> * For FS-NYC and Yahoo-M, all neural tensor models and multi-linear tensor models show competitive performance to each other.
> These two tensors are highly sparse among all the datasets we had, which indicates
> their little information seems to be noisy and makes it difficult for tensor models to capture non-linear or linear patterns.
> However, NeAT shows strong generalization results, even exhibiting the biggest performance gap in Yahoo-M.
>
> * Especially, CoSTCo generally performs better than NCF and NTM due to CNNs, which learn informative patterns using convolutional filters, which are much smaller than heavily parameterized MLPs.
> According to the study [1], MLP-based neural tensor models are prone to overfitting sparse tensors due to MLP's excessive over-parameterization in the form of redundant connections.
> However, NeAT avoids the issue of overparameterization of learning spurious correlations between latent dimensions, via additive components with shallow MLPs, and in doing so achieves the best generalization performance over all baselines for sparse tensors.
>
>
> [1] Liu, Hanpeng, et al. "Costco: A neural tensor completion model for sparse tensors." Proceedings of the 25th ACM SIGKDD International Conference on Knowledge Discovery & Data Mining. 2019.

---

> ### Author Response · Authors · 2023-11-22
> **Responses to Questions**
>
> > How many extra parameters are introduced by NEAT when comparing with CP when the factors CP rank are the same? Could it be the case that, for some sparse tensors, the total number of parameters of NEAT is more than the number of observed entries?
> * We use large-scale sparse tensors for evaluation.
> In this setting, the total number of parameters is more than the number of observed entries even for CPD due to the parameters of factor matrices.
>     * For example, for DBLP (4,057x14,328x7,723), CPD trained with rank 8 has parameters for factors equal to 208,864, which is bigger than nonzeros 150,435 (even if it includes negative samples for training data).
>
> >In that case, how does NEAT do in terms of overfitting?
> * We have mentioned overfitting issues in Section 3.2 in the revision.
> * NeAT jointly trains factor matrices and individually parameterized neural networks, which potentially leads to certain components being in focus for training while others are not.
> Therefore, we apply dropout on all final outputs of MLPs, which randomly selects subsets of components to reconstruct a tensor. This ensures all components can be trained evenly.  Figure 5 (b) exhibits this trend. Additionally, We normalized inputs of MLPs and added weight decay for stable training.
>
> > Does the proposed model work for non-sparse tensors?
> * NeAT can also work for dense tensors the same way it worked for observed entries of the sparse tensor [1, 2]
> * However, the design of NeAT is specialized for sparse tensors, as mentioned earlier in the answer to Weakness 1.
> * Note that recently the vast majority of applications where CPD and neural tensor models have been traditionally successful are on large-scale sparse tensor data (e.g., graphs, user-item interactions, electronic health records, and so on).
> For that reason, we chose to focus on sparse tensors where NeAT can contribute as well.
>
>
> [1] Nickel, Maximilian, Volker Tresp, and Hans-Peter Kriegel. "A three-way model for collective learning on multi-relational data." Icml. Vol. 11. No. 10.5555. 2011.
>
> [2] Oh, Sejoon, et al. "High-performance tucker factorization on heterogeneous platforms." IEEE Transactions on Parallel and Distributed Systems 30.10 (2019): 2237-2248.

---

> ### Author Response · Authors · 2023-11-23
> **Thank you**
>
> Thank you for your constructive comments and valuable insights. They were significantly helpful in improving the presentation of NeAT by making authors think about the model from different perspectives. In the revision, we have tried to further strengthen the analysis of the evaluation and present it clearly, and hope that it will help you understand the empirical results. Also due to limited time, authors tried to desrcribe responses as self-contained as possible  to consider the reviewers time;  we also explained which parts are changed in the revision and summarized how we addressed the reviewers main conern in the general responses.

---

### Official Review · Reviewer_XfPM · 2023-11-01

**Soundness:** 4 excellent
**Presentation:** 4 excellent
**Contribution:** 4 excellent
**Rating:** 8
**Confidence:** 4

**Summary:**

The article presents a new tensor decomposition method called Neural additive tensor decomposition (NEAT), which extends the standard CP Decomposition with non-linear functions. The method incorporates ideas from neural tensor models, and applies Multi-layer perceptrons (MLPs) to each rank-1 CP factors. The proposed method captures non-linear interactions and also helps in interpretability of results.  Numerical results are presented on different datasets to illustrate the performance of the proposed method in comparison to SoTA neural and other tensor decomposition methods.

**Strengths:**

Strengths:
1. The paper presents an interesting new tensor decomposition that captures non-linear interactions
2. The decomposition appears to have an easily interpretable form, and this advantageous in many applications
3. The pair presents extensive numerical results, which show that the proposed outperforms other compared methods.

**Weaknesses:**

Weakness:
From a numerical computation perspective,
1.  The computational cost of the method could be an issue for large tensors.
2. The decomposition is non-unique and the optimization problem looks difficult to solve for a good minima.

**Questions:**

The paper presents an interesting new tensor decomposition, which has several advantages and will likely be useful in a number of applications involving tensors. The paper is well written. Numerical experiments section is extensive and presents many different results illustrating the superior performance of the prosper method, and studies different aspects of the method.

I have the following few minor questions:

1.  How is the optimization problem m in eq (8) computed? Is it just by ADAM or autograd?

2. Movielens seems to be a very easy datasets for neural methods. They all achieve very high accuracy. Is there a particular reason for this?

3. How is the downstream task performed (transductive and inductive) for CP and Tucker decompositions? These do not have parameters to train (and freeze). Are the factor matrices simply used as input features to the classifier?

---

> ### Author Response · Authors · 2023-11-22
> **Responses to Weaknesses**
>
> > The computational cost of the method could be an issue for large tensors.
> * As mentioned in Section 3.3, NeAT scales linearly to the number of given tensor entries rather than the outer dimensions. Generally, real-world sparse tensors present that the number of observed entries is much smaller than their sizes. Therefore, even if tensors have large sizes, if the number of entries is not large, NeAT can efficiently train on those datasets. However, if tensors are dense, this approach can have the same computational cost as if it was dealing with full-size components, as would any other tensor decomposition like CPD or Tucker.
> * Except for nonzeros, the computational complexity is directly affected by the neural network complexity (architecture) and the rank size. According to Tables 2 and 5, and Figures 5 (a) and 7, NeAT performs well on the largest tensor Yahoo-M with a smaller rank size and shallow networks while it requires a larger rank size and a deeper network in the smallest dataset (MovieLens).
> * Large-scale datasets might affect the computation cost of NeAT. However, the time complexity depends more on the characteristics of the datasets. In other words, the number of non-zeros and the MLP network complexity and rank-size required for the tensors are more important than the tensor mode size.
>
> > The decomposition is non-unique and the optimization problem looks difficult to solve for a good minima.
> * This is a valid point. In this paper, our primary goal is to introduce this model and showcase its efficacy and utility and for that reason, we optimize the model with a gradient descent approach. We empirically demonstrate that the convergence of the model is relatively smooth in Figure 7, which points to a well-behaved optimization broadly speaking, as shown by a good generalization of the model when compared to other baselines. We reserve a deeper study on the optimization landscape and different optimization algorithms for this model as future work, while we recognize that it is a very challenging and important topic.

---

> ### Author Response · Authors · 2023-11-22
> **Responses to Questions**
>
> > How is the optimization problem m in eq (8) computed? Is it just by ADAM or autograd?
> * We did not mention how we optimized the model in the previous draft. We mention this in the second paragraph of Section 3.2 of the revised version.
> We train NeAT with Adam using backpropagation. We implemented the model with the Pytorch library.
>
> > Movielens seems to be a very easy datasets for neural methods. They all achieve very high accuracy. Is there a particular reason for this?
> * According to Table 2, neural tensor models with lower ranks show higher accuracy than multi-linear tensor models for the MovieLens dataset. This indicates that movies may not belong to only a linear subspace w.r.t. users and hence neural networks perform well by learning a more non-linear function between movies and users. Additionally, the dataset is relatively smaller and less sparse.
> * We newly report the size and sparsity of tensors and summarize the statistics of them in Table 4.
>
> > How is the downstream task performed (transductive and inductive) for CP and Tucker decompositions? These do not have parameters to train (and freeze). Are the factor matrices simply used as input features to the classifier?
> * Yes factor matrices are used as input features for a downstream task. We consider factor matrices as parameters for CPD and Tucker decomposition models. Therefore, we freeze all factor matrices except for test factors (embeddings) corresponding to new authors.
> * We added an explanation in Section 4.3.1 for clarity.

---

> ### Author Response · Authors · 2023-11-23
> **Thank you**
>
> Thank you for your valuable time and effort to write a review. We also appreciate that the reviewer found this method interesting. It has been an enjoyable journey for the authors developing this method. In the revision, we have tried to further strengthen the analysis of the evaluation and present it clearly, and hope that it will help you better understand the empirical results.

---

> ### Comment · Reviewer_XfPM · 2023-12-01
>
> I thank the authors for their response. I think the paper is solid. However, other reviewers' concern seem valid. I am keeping my score

---

### Author Response · Authors · 2023-11-22
**General Responses**

We would like to sincerely thank the anonymous reviewers for their constructive comments and valuable insights. Your comments were significantly helpful in improving the quality of the paper and in presenting the proposed method clearly.

We have incorporated your requests and updated the draft with an exhaustive and improved array of experiments and explanations. Also, we would like to apologize for the delay in our response as incorporating those changes took much longer than desirable given the short time window of the author's response phase. Our sincerest apologies for the inconvenience, we hope that with those changes we have addressed most of your concerns and can still pursue a short fruitful discussion with you given the time frame.

We carefully and significantly revised our paper to incorporate feedback from all the reviewers. Please see our revised submission, where we have highlighted all major changes in **Blue** color. It would be more coherent to read the changes in the paper but as a handy reference, we summarize the main changes as follows.

* **Difference between neural tensor models and NeAT**: Neural Tensor models paragraph in Section 2
* **Method**
    * **A computational challenge in neuralizing components**: The first paragraph in Section 3.1.
    * **Model’s design choice to improve generalization**: The third and fifth paragraphs in Section 3.1.
    * **Overfitting issue**: The second paragraph in Section 3.2
* **Empirical evaluation**
    * **Link prediction**:
        * In Section 4.2 and Table 2 -- Re-arranged old results, and added a more exhaustive analysis relative to model size.
        * In Appendix 4 and Tables 6 to 8 -- Added new experiments on data sparsity analysis.
    * **The usefulness of components**: Section 4.3 and Figure 3 -- Added explanations on embedding quality of various models.
    * **Sensitivity to hyper-parameters**:
        * In Section 4.4 and Figure 4 -- Re-arranged old results.
        * In Appendix 6 and Figure 8 -- Re-arranged old results.
    * **Parameters usage**: Appendix 3 Table 5 and Figure 6

---

> ### Author Response · Authors · 2023-11-23
> **Main concerns**
>
> We also briefly explained the main concerns reviewers raised and how we addressed them in the revised paper. There are two main concerns as follows.
> * **Lack of the experimental results analysis (R2-W2a, R3-W3, R4-W)**
>     * ***How can NeAT generalize well over baselines?***
> * **Applicability/Effectiveness of Interpretability (R2-W2, R3-W2, R4-S1)**
>     * ***How effective is CPD-like interpretability for real-world applications?***
>
> We have already addressed all concerns and reflected them in the revised paper, which we briefly explain here for convenience.
>
> **Performance of NeAT**
>
> There are two main reasons why NeAT shows superior generalization over baselines:
> 1. **Model design.**
>     * NeAT is designed to additively reconstruct a tensor with individually and separately parameterized components (MLPs and factors).
>     Each component serves as an individual model, which allows NeAT to capture various patterns and learn complex interactions between them.
>     * Also, additive components do not directly interact with each other, unlike all neural tensor models that design all latent dimensions to directly interact, possibly making them learn spurious correlations
>     between various latent dimensions. These components make NeAT less likely to learn spurious correlations between components, preventing overfitting.
>         * Note that by “spurious correlations”, we refer to trained correlations between components to fit the data at the expense of the model's generalization capacity.
>     * Tables 2, 6, 7, and 8 and Figures 3 and 5 (a) demonstrate the results. They consistently show that NeAT outperforms all models when the rank sizes increase, which indicates that NeAT is able to express the tensors accurately using more additive components rather than overfitting.
>
> 2. **Training.**
>     * Applying regularization techniques suitable to the model design of NeAT effectively works and prevents overfitting. Applying dropout on all final output of MLPs helps to train all components appropriately rather than only specific components being used to reconstruct the tensor.
>     * Figure 5 (b) shows that a higher dropout ratio significantly improves NeAT performance. This indicates that even if we use more components (higher rank) to fit the data, NeAT does not easily overfit the data but generalizes the data well.
>
> **Interpretability of NeAT**
> 1. **Importance of Interpretability in Tensor Models.**
>     Tensor decomposition models have attracted attention not only for their effectiveness in tensor completion tasks but also for their interpretability. Unlike, say, autoencoder-based representations, what is special about tensor factor matrices (especially from the CPD model) is that they are a compact representation of the unique hidden patterns in the data, and we can directly interpret them by looking at the factor coefficient values, and understand those patterns by mapping those factors to the original data. As a result, tensor models are significantly useful for users who want to explore underlying patterns.
>     * For example, Interpretability is essential for healthcare analysis. Electronic health records (EHR) can be represented as a tensor that consists of (patients, medical features, time). Tensor models can help healthcare professionals to understand the relations between patients and medical features [1].
> 2. **Benefits of Additivity (Simplicity) for Interpretability.**  Additive components from CPD and NeAT allow users to discover the hidden patterns easily rather than interlinked components (Tucker or other neural tensor models). When we analyze factors, we need to consider the interactions between components for accurate analysis, which complicates the process of discovering patterns. When the rank size increases, we need to consider R^N (Rank^Mode) possibilities, which are computationally very expensive.
> 3. **Comparison to Tucker Interpretability**
>     Tucker interpretability is used when the focus is to know the correlations between components (which requires more complicated analysis). However, in those cases, a dense core is not especially useful and therefore sparsity constraints need to be imposed on the core. In those cases, where the number of component interactions is small, we can rewrite that model as a CPD model (with the extreme case of a superdiagonal core, which can be written exactly as a CPD model).
>
> [1] Ho, Joyce C., Joydeep Ghosh, and Jimeng Sun. "Marble: high-throughput phenotyping from electronic health records via sparse nonnegative tensor factorization." Proceedings of the 20th ACM SIGKDD international conference on Knowledge discovery and data mining. 2014

---

### Meta-Review · Area_Chair_qkts · 2023-12-10

**Metareview:**

The paper proposes a neural additive tensor decomposition: sum of r terms, each generalizes rank-1 tensors.
The decomposition is then compared to other tensor models.

Strengths:
1) The idea is potentially interesting
2) The results comparing to classical tensor models show some improvement for tensor completion

Weaknesses:
1) No theoretical analysis or motivation, at least for the generalization ability of the work.
2) The paper claims 'interpretability' with respect to neural networks, which is also not clear
2) No comparison with other deep tensor models.

To summarize, deep tensor/hybrid model should solve some problem that clearly can not be solved either by tensor approach or by neural networks adequately. Just introducing a model and counting parameters is not enough. A simple model example is necessary.

**Justification For Why Not Higher Score:**

The paper is not very motivating and 'deep': lets build a model and compare on tensor completion. Why it is better than previous methods? Why it should be preferred to neural networks or tensors? Not cnvincing.

**Justification For Why Not Lower Score:**

N/A

---

### Decision · Program_Chairs · 2024-01-16

Reject